# The type of DNA damage response after decitabine treatment depends on the level of DNMT activity

Tina Aumer[1,2,*], Maike Däther[1,2,*], Linda Bergmayr[2,*], Stephanie Kartika[3], Theodor Zeng[2], Qingyi Ge[2], Grazia Giorgio[4], Alexander J Hess[2], Stylianos Michalakis[4], Franziska R Traube[1,2,5]

Decitabine and azacytidine are considered as epigenetic drugs that induce DNA methyltransferase (DNMT)-DNA crosslinks, resulting in DNA hypomethylation and damage. Although they are already applied against myeloid cancers, important aspects of their mode of action remain unknown, highly limiting their clinical potential. Using a combinatorial approach, we reveal that the efficacy profile of both compounds primarily depends on the level of induced DNA damage. Under low DNMT activity, only decitabine has a substantial impact. Conversely, when DNMT activity is high, toxicity and cellular response to both compounds are dramatically increased, but do not primarily depend on DNA hypomethylation or RNA-associated processes. By investigating proteome dynamics on chromatin, we show that decitabine induces a strictly DNMT-dependent multifaceted DNA damage response based on chromatin recruitment, but not expression-level changes of repair-associated proteins. The choice of DNA repair pathway hereby depends on the severity of decitabine-induced DNA lesions. Although under moderate DNMT activity, mismatch (MMR), base excision (BER), and Fanconi anaemia–dependent DNA repair combined with homologous recombination are activated in response to decitabine, high DNMT activity and therefore immense replication stress induce activation of MMR and BER followed by non-homologous end joining.

## Introduction

5-Aza-2′-deoxycytidine (decitabine, AzadC) and 5-azacytidine (Azacytidine, AzaC) are cytosine analogues that covalently trap DNA methyltransferases (DNMTs) and therefore belong to the compound class of hypomethylating agents (HMAs) (1, 2). Both compounds are applied in the clinic against myelodysplastic syndrome and acute myeloid leukaemia (AML), which have otherwise very limited treatment options (3, 4). It has been reported that AzadC and AzaC have potentially beneficial effects for therapy of solid tumours

as well, but it is not understood yet why solid tumours do not equally respond towards AzadC or AzaC exposure as haematopoietic malignancies (5, 6, 7, 8). AzaC and AzadC feature a multi-mode of action (multi-MoA) by addressing epigenetic and DNA damage processes, and in addition for AzaC, also RNA-dependent processes (Fig 1A) (1, 9). Both compounds are taken up into cells via nucleoside transporters in the plasma membrane, however with different transportability profiles with respect to the different nucleoside transporter types (10). After uptake, 80–90% of AzaC is incorporated into RNA, where it affects nucleic acid and protein metabolism by destabilizing RNA. Among other effects, AzaC inhibits the tRNA (cytosine[38]-C[5]) methyltransferase (*TRDMT1*, *DNMT2*) and reduces protein expression levels of ribonucleotide reductase subunit 2 (RRM2) in blood and leukaemic cells. The ribonucleotide reductase converts ribonucleotides to 2′-deoxyribonucleotides for DNA synthesis, and the consequently lower protein levels of RRM2 after AzaC treatment were found to be an important mechanism by which AzaC blocks cell proliferation in leukaemic cells (9, 11, 12, 13). The remaining 10–20% of AzaC are converted on the diphosphate level to the respective AzadC analogue, and the metabolic pathways of AzaC and AzadC unite (Fig 1A). The AzadC triphosphate can be subsequently used by DNA polymerases for genomic incorporation of AzadC during the S phase instead of 2′-deoxycytidine (dC). After genomic incorporation, AzadC is recognized by DNMTs as dC, but because of the CH-N replacement at the pyrimidine ring, DNMTs cannot be released anymore after the nucleophilic attack, resulting in permanent crosslinks between the protein and the 5-aza-cytosine nucleobase (1) (Fig 1A). On the epigenome level, DNMT inhibition leads to a global loss of the epigenetic mark 5-methyl-2′-deoxycytidine (mdC), a key player of epigenetic modulation of gene expression (14). This feature can be highly beneficial for tumour therapy as many cancer types and in particular AML subtypes have silenced tumour suppressor genes by hypermethylation of the respective promoter regions (15, 16, 17, 18). On the DNA damage level, AzadC can potentially create various types of DNA lesions (Fig 1A). First, AzadC has a non-canonical base, which promotes rapid

[1]Institute of Chemical Epigenetics Munich, Department of Chemistry, University of Munich (LMU), München, Germany   [2]TUM School of Natural Sciences, Technical University of Munich (TUM), München, Germany   [3]Department of Biochemistry, University of Munich (LMU), München, Germany   [4]Department of Ophthalmology, University Hospital LMU Munich, München, Germany   [5]Institute of Biochemistry and Technical Biochemistry, University of Stuttgart, Stuttgart, Germany

Correspondence: franziska.traube@ibtb.uni-stuttgart.de
*Tina Aumer, Maike Däther, and Linda Bergmayr contributed equally to this work

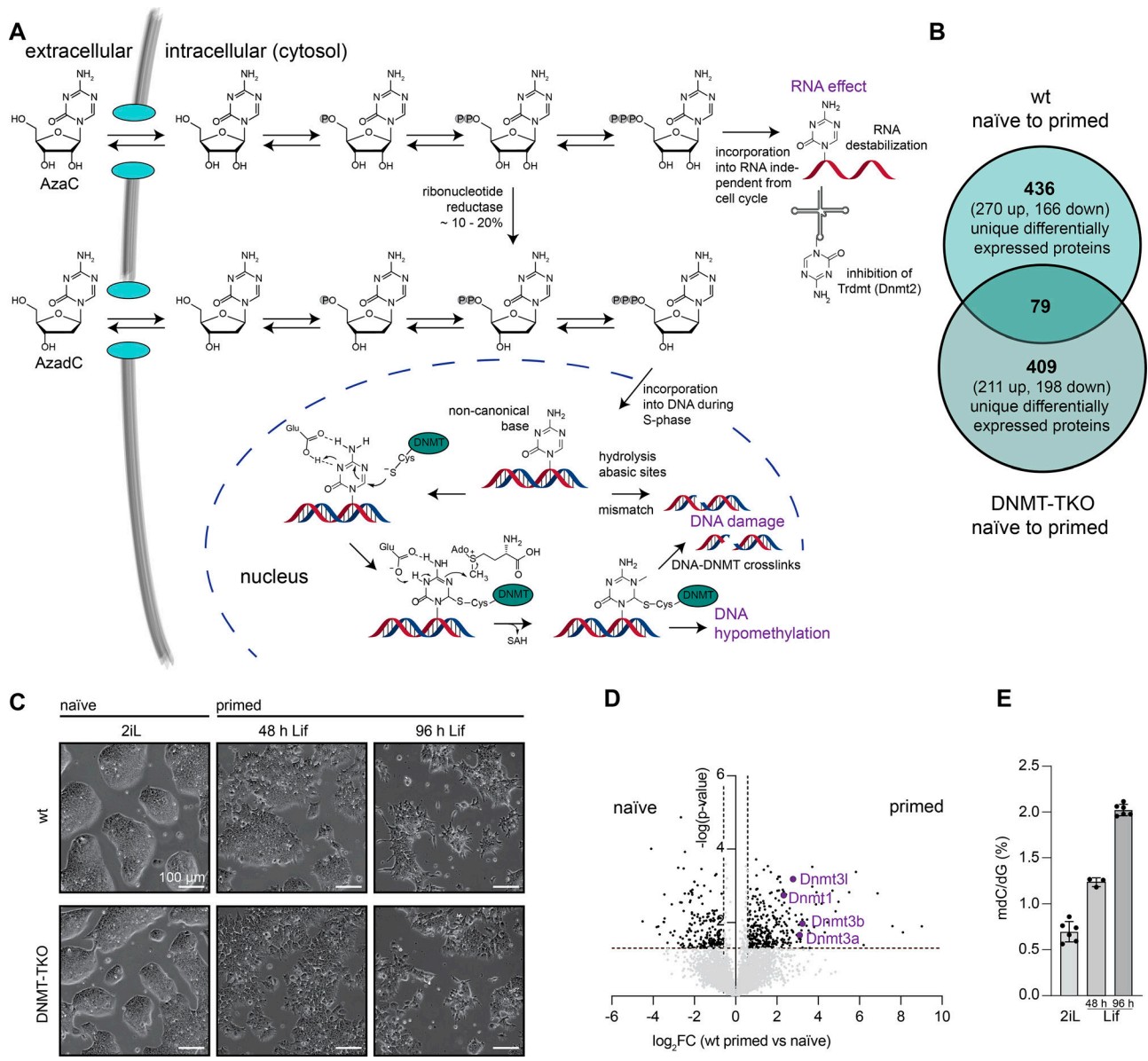

**Figure 1. Mouse embryonic stem cells (mESCs) as a model system to study the different modes of action (MoAs) of AzadC and AzaC.**
**(A)** Schematic representation of the metabolization of AzadC and AzaC in the cell. **(B)** Venn diagram of the number of proteins in the wt and the DNMT-TKO that showed significant expression-level changes ($|log_2FC| \geq 0.58496$ and $P < 0.05$) from the naïve to the 96-h primed state. 79 common proteins were significantly differentially expressed in both genotypes. **(C)** Morphological changes as indicated by representative brightfield microscopy of wt and DNMT-TKO mESCs in the transition from the naïve state (cells cultured in 2iL) to the primed state (cells cultured in Lif). **(D)** Volcano plot showing the protein expression changes ($log_2FC$) and the consistency of the change ($-log[P$-value]) between the naïve and the primed state in the wt. Proteins with higher expression in the 96-h primed state are displayed on the right side, and proteins with higher expression in the naïve state are shown on the left side. Dnmt1, Dnmt3a, Dnmt3b, and the DNMT3-regulatory unit Dnmt3l are displayed in purple. Proteins with significant expression changes ($|log_2FC| > 0.58496$, $-log[P$-value] > 1.3 [≙ $P < 0.05$]) are shown in black, and the rest is shown in grey. **(E)** Quantification of absolute 5mdC in the DNA by triple-quadruple mass spectrometry, normalized to the amount of dG.

hydrolysation of the nucleoside (19), resulting in DNA lesions by mismatches and abasic sites. However, the most severe form of DNA damage after exposure to AzadC, which also determines the mutagenicity of AzadC, is the DNMT-DNA crosslinks (20, 21, 22).

To maintain a level of genomic integrity that is necessary to survive, cancer cells often have a very efficient, but less precise, DNA repair machinery (23). This feature gives them a survival advantage not only to deal with naturally occurring DNA lesions, but also to deal with DNA lesions introduced by DNA-damaging reagents (24). One clinically important example of chemoresistance by proficient DNA repair is the on-target resistance of many aggressive tumours against cisplatin, which is applied as a cytostatic agent against solid tumours (25). To understand the MoA and the associated repair mechanisms of DNA-damaging reagents is therefore of utmost importance to improve therapy efficiency in the clinic. Previous studies showed that non-homologous end joining

(NHEJ), homologous recombination (HR) via the Fanconi anaemia (FA) pathway, and poly(ADP-ribose) polymerase 1 (PARP1)–dependent DNA repair are involved in the repair of AzadC-induced DNA lesions (26, 27, 28). However, it has not been investigated in a holistic and systematic manner yet which DNA damage responses (DDRs) are activated as a cellular response towards AzadC or AzaC under different cellular prerequisites. This information is pivotal to understand and break resistance mechanisms during AzadC- or AzaC-based cancer therapy.

The multi-MoA profiles of AzadC and AzaC provide unique drug profiles in comparison with other chemotherapeutic agents that target either genomic integrity or other cellular features such as epigenome patterns. However, to date it is unclear how much the individual MoAs contribute to the efficacy profile of AzadC and AzaC and reliable biomarkers to predict a patient's response to both, exclusively one, or none of the two compounds are still missing. To investigate in detail how much the different MoAs of AzadC and AzaC contribute to their efficacy profile, we chose mouse embryonic stem cells (mESCs) as a model system because their special cellular features allowed us to distinguish the different MoAs from each other (Table 1). In contrast to other cell types, mESCs in a non-committed state can tolerate ablation of Dnmt1, Dnmt3a, and Dnmt3b (DNMT-TKO) (Fig S1A and B), resulting in global DNA demethylation (Fig S1C and D) without showing deleterious developmental and cellular defects (29). Furthermore, mESCs also show sustained proliferative signalling and replicative immortality like cancer cells, but do not feature genomic instability. Importantly, they have a fully functional DDR, which makes them exceptionally suitable to study potential proliferation inhibitory and cell death–inducing effects of AzadC and AzaC, as well as involved DNA repair processes. Last, mESCs can be cultured in a naïve (2iL culture conditions) or in a primed (Lif culture conditions) state, which represents two pluripotent but distinct developmental stages, where the naïve state is characterized by a very low, but the primed state by a very high, DNMT activity (30, 31).

To dissect the individual contribution of the different MoAs and gain a holistic picture of associated DNA repair mechanisms, we started with AzadC treatment of the wt and the DNMT-TKO in the naïve state to distinguish DNMT-dependent effects from other DNA–based effects. Next, we studied the effects of AzadC on both genotypes in the primed state when DNMT activity is high compared with the naïve state. Last, we had a closer look at the effects of AzadC in the naïve and the primed state in both genotypes and also investigated the effects of DNMT inhibition without creating DNA damage using the previously reported non-nucleoside DNMT inhibitor RG108 (32, 33).

## Results

### mESCs as a model system to study the mode of action of AzadC and AzaC

Before we started to investigate the effects of AzadC and AzaC using our anticipated model system, we checked whether wt and the DNMT-TKO cells provided similar cellular features, for example,

distinct morphological changes upon priming, despite the absence of all three DNMT enzymes in the DNMT-TKO. Upon priming, a comparable number of proteins showed significant expression-level changes from the naïve to the primed state for both genotypes (Fig 1B). The differently expressed proteins between the naïve and the primed state differed, however, substantially between the two genotypes. Nevertheless, both genotypes underwent similar morphological changes. When cultured in 2iL, both wt and DNMT-TKO cells formed densely packed colonies with distinct borders. Within the colonies, individual cells could not be spotted (Fig 1C, naïve). Upon priming, the colonies dissected and individual cells with distinct morphology were increasingly observed for both genotypes (Fig 1C, primed). Moreover, in the wt, Dnmt1, Dnmt3a, and Dnmt3b were all significantly more expressed in the 96-h primed state compared with the naïve state (Fig 1D), which consequently resulted in higher DNMT activity as indicated by increasing levels of 5mdC (Fig 1E). Because new 5mdC patterns are established during priming, wt mESCs allow to monitor closely the effects of AzadC and AzaC when DNMT activity is highly dynamic in cells with the same genetic background. Altogether, these results indicated that the wt and the DNMT-TKO mESCs undergo fundamental cellular changes from the naïve to the 96-h primed state. Although the individual changes on the proteome level were different between the two genotypes, the extent and the resulting morphological changes were similar, and therefore, we decided that the mESCs provided a suitable model system to directly compare the effects of AzadC and AzaC in the presence and absence of DNMT enzymes.

### Presence of genomic AzadC substantially contributes to the anti-proliferative and cytotoxic effect without formation of DNMT-DNA crosslinks when overall DNMT activity is low

To investigate how much the presence of genomic AzadC contributes to its toxicity by base mismatch (MoA 3) or formation of abasic sites by spontaneous hydrolysis of the base (MoA 4), independent from DNA hypomethylation (MoA 1) and DNA damage by DNMT crosslinking (MoA 2), we first compared the effects of AzadC on wt and DNMT-TKO cells under naïve conditions (2iL) when DNMT expression and activity are low in the wt (Fig 1D and E). First, we confirmed by our previously reported QQQ-MS method for exact quantification of nucleosides (34) that AzadC had the anticipated hypomethylating effect in the wt even under low DNMT activity (Fig 2A). We detected a significant decrease in the amount of mdC after 48-h treatment with 1.25 or 2.5 $\mu$M of AzadC with no difference between the concentrations, indicating that the hypomethylating effect was already at the maximum at the concentrations applied. Because the mdC levels of the DNMT-TKO were under the limit of detection even without treatment (Fig S1D), a DNA hypomethylating effect could not be observed upon treatment. Even though DNA demethylation was already at the maximum at 1.25 $\mu$M AzadC in the wt, inspection of the phenotypic changes of the cells by brightfield microscopy revealed that the anti-proliferative and cytotoxic effect of AzadC was strictly concentration-dependent with only mild effects at lower concentrations (Fig 2B). This result was in line with previous reports that AzadC-induced DNA hypomethylation already occurs at low concentrations before inducing cytotoxic effects (9, 35). Interestingly, the DNMT-TKO showed a very similar cell viability

**Table 1. Modes of action of HMAs and how they can be distinguished from each other.**

| Mode of action (MoA) | Conditions in which the MoA is present | Conditions in which the MoA is the only or the dominant MoA compared with other conditions |
| --- | --- | --- |
| 1. DNA hypomethylation | wt with AzadC, AzaC, and RG108 | Only MoA in RG108-treated wt cells |
| 2. DNA damage by DNA-DNMT crosslinks | wt with AzadC and AzaC | Dominant MoA in AzadC-treated wt cells when effects of MoA1 are subtracted |
| 3. & 4. DNA lesions by mismatch and formation of abasic sites | wt and DNMT-TKO with AzadC and to a much lesser extent (10–20%) with AzaC | Only MoA in AzadC-treated DNMT-TKO cells |
| 5. RNA-dependent effect | wt and DNMT-TKO with AzaC | Dominant MoA in AzaC-treated DNMT-TKO cells |

RG108 is a non-nucleoside–based DNMT inhibitor.

pattern in the brightfield microscopy images, albeit starting at higher concentrations (Fig 2B). We confirmed this observation in an independent flow cytometry–based apoptosis assay, where we observed a concentration-dependent increase in apoptotic events for both genotypes (Fig 2C). Compared with the respective un-treated controls, the increase in apoptotic events was significant from 2.5 $\mu$M of AzadC on in the wt and from 5 $\mu$M of AzadC on in the DNMT-TKO (Fig 2D). Next, we compared the proliferation rate of the wt and the DNMT-TKO. Untreated controls showed an identical proliferation rate with a doubling time of 17–19 h. Treatment with AzadC slowed down proliferation in both genotypes with a higher impact on the wt (Fig S2A). However, proliferation rates after treatment were overall within the same range for both genotypes, and after treatment with 2.5 $\mu$M AzadC, proliferation was slowed down in both genotypes to the same extent with a doubling time of ~26 h (Fig 2E).

The similarity of phenotypic (Fig 2B–D) and proliferative (Figs 2E and S2A) changes in the wt and DNMT-TKO after 48-h treatment with AzadC was unexpected because AzadC cannot induce deleterious DNMT-DNA crosslinks in the DNMT-TKO and should therefore have a substantially lower impact in the DNMT-TKO on cellular well-being. There were two possible options to explain those results. Either AzadC was metabolized differently in the wt and the DNMT-TKO, with higher uptake and genomic incorporation in the DNMT-TKO, or AzadC itself without formation of DNMT-DNA crosslinks contributed substantially to the anti-proliferative and cytotoxic effects at the concentrations applied. To test which option explained our results, we checked AzadC uptake and genomic incorporation in the wt and the DNMT-TKO by treating both genotypes with 2.5 $\mu$M of AzadC for 24 and 48 h and quantifying the global, sequence context-independent, genomic incorporation levels of AzadC by QQQ-MS (Fig 2F). For the 24-h timepoint, we added 2.5 $\mu$M of AzadC once at the start (0 h) and harvested the cells after 24 h (treatment a). For the 48-h timepoint, we treated the cells for 48 h before harvest, but tested three different treatment regimens—addition of 2.5 $\mu$M of AzadC at 0 h and neither medium change nor second compound addition (treatment b); addition of 2.5 $\mu$M of AzadC at 0 h and medium change after 24 h but no second compound addition (treatment c); and addition of 2.5 $\mu$M of AzadC at 0 h and medium change after 24 h with second compound addition of 2.5 $\mu$M of AzadC (treatment d). The untreated control served as a background control, and as expected, an AzadC signal was not detectable neither in the wt nor in the DNMT-TKO (Fig S2B). For all timepoints and treatment regimens tested, we did not observe any difference

between the wt and the DNMT-TKO (Fig 2F), indicating that there was no difference between the two genotypes regarding AzadC uptake and metabolization. However, we detected substantial differences between the different treatment regimens. After 24 h, the amount of genomically incorporated AzadC, normalized to the amount of dG, was about two times higher compared with the 48-h timepoint when no additional compound was added. In contrast, when AzadC was added twice (after 0 h and in addition after 24 h), we detected an additional increase in genomic AzadC after 48 h compared with the 24-h timepoint. These results suggested that 24 h after addition, all available AzadC had been either incorporated or inactivated by hydrolysis or removal from the genome. Taken into account that the proliferation rates of the wt and the DNMT-TKO cells were close to 24 h when exposed to 2.5 $\mu$M of AzadC, these results implied that removal of genomically incorporated AzadC mostly depended on the global scale on passive dilution by cell division (Fig 2G), whereas removal by repair mechanisms or spontaneous decay only played a minor role. In consequence, to avoid passive dilution and to in-crease the amount of genomically incorporated AzadC, the com-pound has to be added continuously (Fig 2H).

To also investigate the formation of DNA damage in the wt and the DNMT-TKO after AzadC treatment in a dose-dependent manner, we performed an immunoblot analysis against $\gamma$H2AX (Figs 2I and S2C). H2AX is a histone variant, which is placed as a mark at sites of DNA double-strand breaks (DSBs) and is sub-sequently phosphorylated at Ser-139 ($\gamma$H2AX) to recruit the repair machinery. Because one $\gamma$H2AX is placed per DSB, it is a very sensitive and quantitative marker for DSB formation (36). As ex-pected, we detected only a very faint $\gamma$H2AX signal in the untreated controls of both genotypes, whereas AzadC treatment with all concentrations tested (1.25–5 $\mu$M) resulted in comparable for-mation of DSBs in the wt. In the DNMT-TKO, we observed strict dose-dependent formation of DSBs with no detectable signal after 1.25 $\mu$M AzadC treatment, but a clearly visible signal at the higher concentrations. In parallel, we detected the levels of Parp1, which serves not only as a marker for DNA damage but also in-duction of apoptosis. Parp1 can be cleaved in a 24-kD fragment, which remains at the lesion, and an 89-kD fragment that is re-leased in the cytosol to induce apoptosis (37). The total Parp1 levels did not differ between the wt and the DNMT-TKO, but with slightly lower levels in the untreated wt (Figs 2I and S2C). In contrast, the 89-kD cleaved fragment showed only a strong signal in the wt, already at a concentration of 1.25 $\mu$M of AzadC, but not in the DNMT-TKO at any concentrations tested.

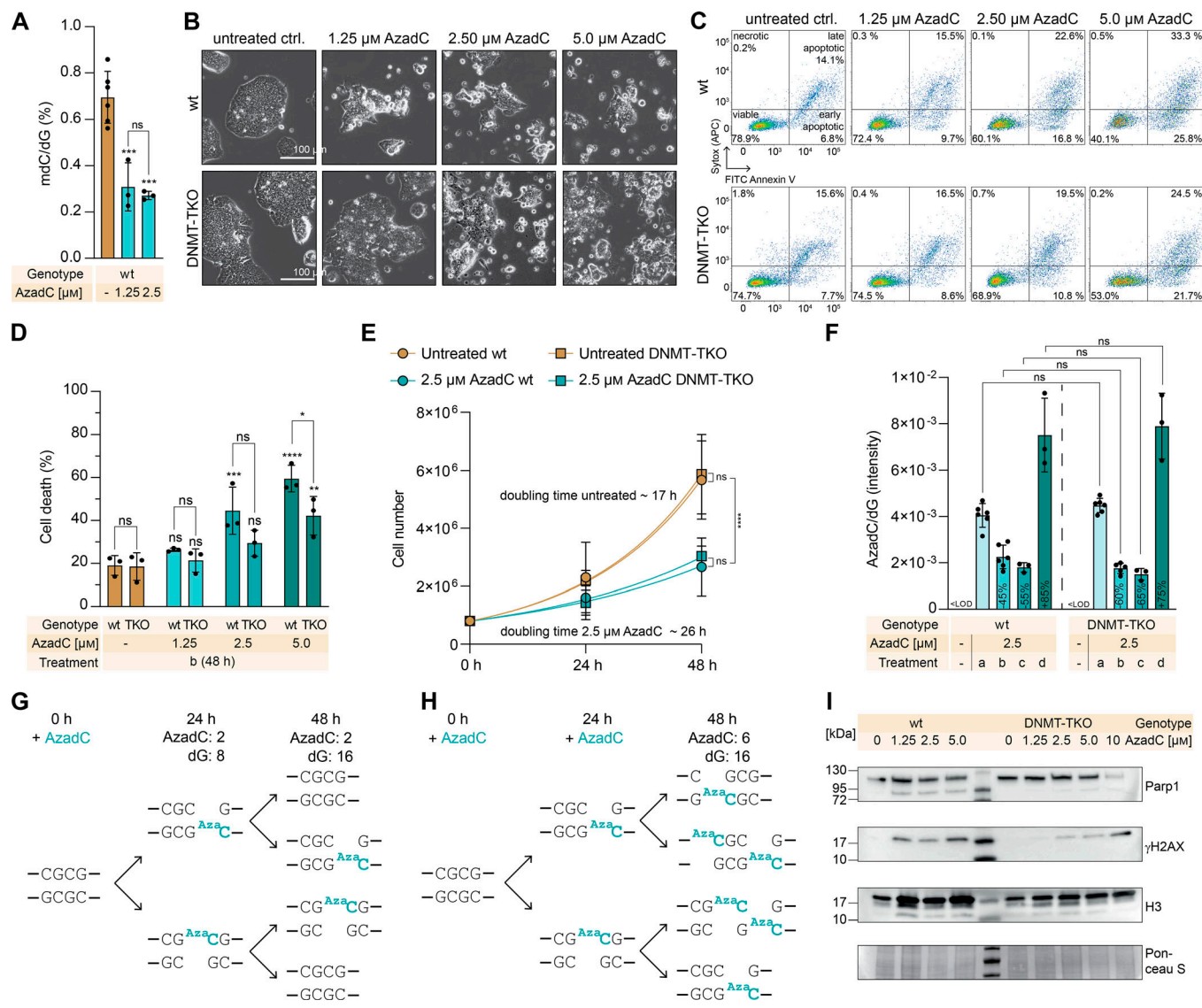

**Figure 2. Effects of AzadC treatment in wt and DNMT-TKO mESCs under 2iL conditions.**

**(A, B, C, D, E, I)** Treatment with AzadC for 48 h with one compound addition at 0 h. **(A, D, F)** Bar represents the mean, error bars represent the SD, and each dot represents one biologically independent replicate. **(A, D, E, F)** ns $P_{adj}$ > 0.05, * 0.05 > $P_{adj}$ > 0.01, ** 0.01 > $P_{adj}$ > 0.001, *** 0.001 > $P_{adj}$ > 0.0001, **** $P_{adj}$ < 0.0001. Stars above bars of AzadC-treated samples indicate a significant difference in the mean compared with the respective untreated control. **(A)** Amount of mdC, quantified by QQQ-MS and normalized to the amount of dG. One-way ANOVA combined with Tukey's multiple comparisons test (Supplemental Data 1, Fig 2A). **(B)** Representative brightfield microscopy images of wt and DNMT-TKO cells after treatment with increasing concentrations of AzadC. **(C)** Representative flow cytometry scatter plots of wt and DNMT-TKO cells after 48-h treatment with increasing concentrations of AzadC compared with the untreated control (n = 10,000 events per condition) using FITC–Annexin V binding as a marker for apoptosis and SYTOX Red Dead Cell Stain as a marker for dead cells. Viable cells are Annexin V⁻/SYTOX⁻, cells in an early apoptotic state are Annexin V⁺/SYTOX⁻, cells in a late apoptotic state are Annexin V⁺/SYTOX⁺, and cells that have been died from other reasons (necrosis, etc.) are Annexin V⁻/SYTOX⁺. **(D)** Summary of cell death events (necrotic + early apoptotic + late apoptotic as indicated by panel (C)) in wt and DNMT-TKO cells after 48-h treatment with AzadC in increasing concentrations compared with the untreated control. Two-way ANOVA (genotype and treatment) combined with Šidák's and Dunnett's multiple comparisons test to compare the same treatment between the two genotypes and to compare the treated samples with the respective untreated control within one genotype (Supplemental Data 1, Fig 2D). **(E)** Proliferation curve of wt and DNMT-TKO cells after treatment with 2.5 µM of AzadC compared with the untreated controls. For each sample to be measured, 800,000 cells were seeded initially (0 h). For the 24-h and the 48-h timepoints, three biologically independent replicates were quantified. The symbol represents the mean, and the error bar represents the SD. Fitting of the growth curve by exponential (Malthusian) growth with the constraint $Y_0$ = 800,000 (Supplemental Data 1, Fig 2E). **(F)** Intensity of AzadC signal normalized to the intensity of dG signal in genomic DNA, measured by QQQ-MS, in the wt and the DNMT-TKO after treatment with 2.5 µM of AzadC. LOD = limit of detection, treatment a = AzadC addition at 0 h, harvest after 24 h; b = AzadC addition at 0 h, harvest after 48 h; c = AzadC addition at 0 h, medium change after 24 h to medium without AzadC, harvest after 48 h; d = AzadC addition at 0 h, medium change after 24 h to medium with freshly added AzadC, harvest after 48 h. Two-way ANOVA (genotype and treatment) combined with Šidák's multiple comparisons test to compare the same treatment between the two genotypes (Supplemental Data 1, Fig 2F). **(G, H)** Schematic representation of how the amount of AzadC in relation to the amount of dG changes when AzadC is only available in the soluble pool for 24 h and the proliferation rate is 24 h. **(I)** Immunoblot analysis of Parp1 and γH2AX levels in wt and DNMT-TKO cells treated with increasing concentrations of AzadC. Histone H3 and Ponceau S staining served as a loading control.

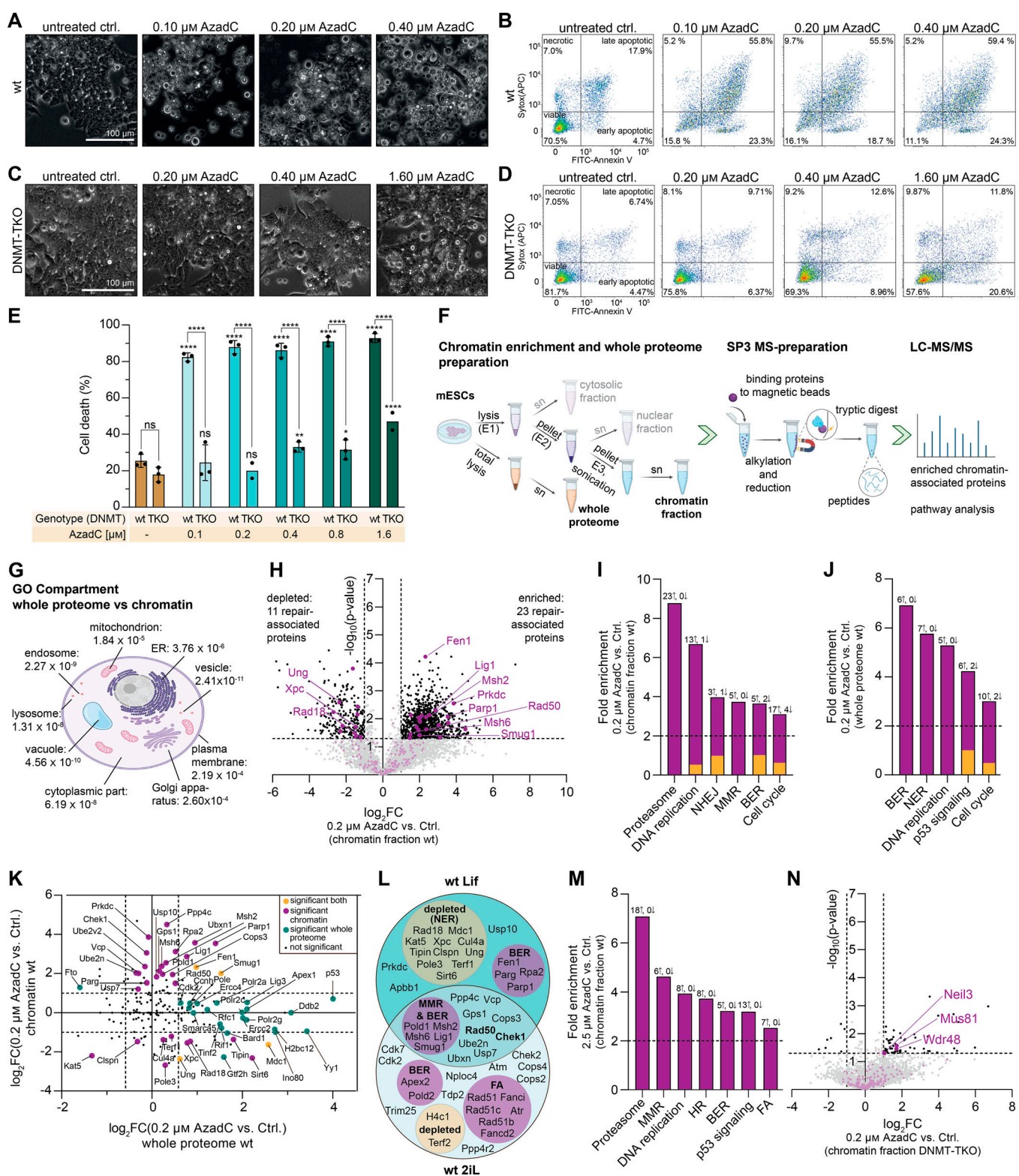

**Figure 3. Effects of AzadC treatment in mESCs under Lif conditions and involved DNA repair pathways under Lif and 2iL conditions.**

Lif conditions are 48-h pre-incubation in Lif medium, followed by 48-h treatment in Lif medium (96 h Lif in total). **(A)** Representative brightfield microscopy images of wt cells after treatment with increasing concentrations of AzadC (Lif). **(B)** Representative flow cytometry scatter plots of wt cells (Lif) with increasing concentrations of AzadC compared with the untreated control (n = 10,000 events per condition) using FITC–Annexin V binding as a marker for apoptosis and SYTOX Red as a marker for dead cells. **(C)** Representative brightfield microscopy images of DNMT-TKO cells after treatment with increasing concentrations of AzadC (Lif). **(D)** Representative flow cytometry scatter plots of DNMT-TKO cells (Lif) with increasing concentrations of AzadC compared with the untreated control (n = 10,000 events per condition). **(B, D, E)** Summary of

In summary, our results suggest that genomically incorporated AzadC is mainly removed by passive dilution during cell division unless a new substance is provided at least every 24 h. AzadC already exhibits an intrinsic anti-proliferative and apoptotic effect independent from DNMT crosslinking, but dependent on active DNA replication. After 24 h, when AzadC concentration was at a maximum (Fig 2D), the cells had undergone one replication cycle, and already a significant reduction of mdC (12), but apoptosis was only induced to a very low extent (Fig S2D). This result was in accordance with previous studies showing that the cytotoxic potential of AzadC unravels during the second DNA replication cycle after incorporation (26).

### When DNMT activity is high, DNA-DNMT crosslinking dominates the efficacy profile of AzadC

Upon priming, wt mESCs undergo reprogramming, including changing and overall increasing DNA methylation patterns in the wt (30, 31) as a result of higher expression of all three DNMT enzymes and higher DNMT activity (Fig 1D and E). Therefore, we expected increased sensitivity of the wt towards AzadC treatment. Whereas AzadC-induced DNA lesions that do not originate from DNA-DNMT crosslinking are strictly dose-dependent as they only depend on the amount of incorporated AzadC, DNA lesions from DNA-DNMT crosslinking depend on the amount of incorporated AzadC but even more on the activity of DNMT enzymes. With increasing amounts of these crosslinks, genome instability is multiplied. Consequently, when DNMT activity is high, a lower amount of genomically incorporated AzadC can nevertheless lead to more devastating effects than a substantially higher amount of AzadC under conditions of low DNMT activity. In line with these preliminary considerations, we did not observe any proliferation of AzadC-treated wt cells under Lif conditions (Fig S3A). Furthermore, brightfield images indicated that under Lif conditions, massive cell death already occurred in the wt when the cells were treated with as little as 0.1 $\mu M$ of AzadC (Fig 3A). The drastic increase in dead cells after low-dose AzadC treatment was in addition confirmed by the flow cytometry–based apoptosis and cell death assay (Fig 3B). In contrast, the DNMT-TKO

did not respond equally sensitive (Figs 3C and S3A). However, primed DNMT-TKO mESCs were also more sensitive towards low concentrations of AzadC (Fig 3D) than the ones under naïve conditions, which did not respond to an AzadC concentration below 2.5 $\mu M$ (Fig 2B). Immunofluorescence analysis of γH2AX in AzadC-treated DNMT-TKO cells under naïve and primed conditions suggested that DNA damage was already present in the primed state at lower concentrations of AzadC (Fig S3B), but the underlying reason was not apparent. Nevertheless, apoptotic events in the wt compared with DNMT-TKO after AzadC treatment were disproportionately higher in the primed state compared with the naïve one (Fig 3E). Overall, these results suggested that the DNMT-dependent effects are dominant for the toxicity profile of AzadC when DNMT activity is high and DNMT-independent DNA lesions only play a subordinate role.

### The activation of specific DNA repair pathways as a response to AzadC-induced DNA lesions depends on DNMT activity

To investigate the DDR towards AzadC in a systematic and holistic way, we treated wt and DNMT-TKO cells with AzadC under Lif and 2iL conditions. Afterwards, we divided the cells into two portions—one to enrich the chromatin fraction to check for recruitment of DNA repair–associated proteins to the site of action, and one to isolate the whole proteome to check for their expression-level changes. Subsequently, the isolated proteins were subjected to an established SP3 workflow and analysed by LC-MS/MS (Fig 3F). To validate our workflow, we first analysed the detected proteins in the chromatin fraction against the whole proteome and vice versa. GO-term analysis revealed that in the chromatin fraction, GO terms associated with regulation of (DNA-templated) transcription (e.g., GO:0006355, GO:0006357), DNA binding (e.g., GO:0003677, GO: 0043565, GO:0000976), and DNA-binding transcription factor activities (e.g., GO:0003700, GO:0000981, GO:0001228, GO:0001227) were highly overrepresented compared with the whole proteome (Fig S3C, Supplemental Data 2). On the contrary, the whole proteome showed massive enrichment for all cellular compartments except the nucleus when compared to the proteins of the chromatin fraction (Fig 3G). The comparison in both directions confirmed that

cell death events (necrotic + early apoptotic + late apoptotic as indicated by panels (B, D)) in wt and DNMT-TKO cells after 48-h treatment with AzadC in increasing concentrations compared with the untreated control. Bars represent the mean, error bars represent the SD, and dots represent biologically independent replicates. Two-way ANOVA (genotype and treatment) combined with Šidák's and Dunnett's multiple comparisons test to compare the same treatment between the two genotypes and to compare the treated samples with the respective untreated control within one genotype (Supplemental Data 1, Fig 3E). Stars above bars of AzadC-treated samples indicate a significant difference in the mean compared with the respective untreated control. ns $P_{adj}$ > 0.05, * 0.05 > $P_{adj}$ > 0.01, ** 0.01 > $P_{adj}$ > 0.001, *** 0.001 > $P_{adj}$ > 0.0001, **** $P_{adj}$ < 0.0001. **(F)** Workflow for chromatin enrichment and whole proteome analysis and subsequent LC-MS/MS measurement. E1–E3 are different buffer formulations that can solubilize different subcellular compartments. **(G)** Significantly enriched Gene Ontology terms (GO Compartment) when the proteins detected in the whole proteome of wt cells were compared with the respective chromatin fraction (Lif conditions). Figure panels (F, G) were created with the help of BioRender.com. **(H)** Volcano plot of chromatin-enriched proteins: after AzadC treatment of wt cells under Lif conditions (left side: untreated control; right side: 0.2 $\mu M$ AzadC-treated mESCs). Not significantly enriched proteins (−log[$P$-value] < 1.3 and |$log_2$FC| < 1) are marked grey. Significantly enriched proteins in one of the two conditions are labelled black. Proteins that are involved in DNA repair according to Reactome are labelled purple. **(H, I)** Pathway enrichment analysis of the data in (H) performed with pathfindR. **(J)** Pathway enrichment analysis of the whole proteome data of AzadC-treated mESCs under Lif conditions compared with the untreated ctrl. using pathfindR. **(I, J)** Number of enriched (I)/up-regulated (J) (↑, purple) and depleted (I)/down-regulated (J) (↓, orange) proteins after AzadC treatment that are assigned to the different pathways are indicated. **(K)** Correlation plot of the expression-level changes (x-axis, whole proteome) or of the chromatin enrichment changes (y-axis, chromatin fraction) of DNA repair–associated proteins of AzadC-treated mESCs under Lif conditions in comparison with the untreated control. **(L)** Venn diagram of significantly chromatin-enriched and chromatin-depleted (marked with an orange circle) proteins after AzadC treatment of wt cells under Lif and 2iL conditions. Proteins that are assigned to a specific DNA repair pathway are grouped. **(M)** Pathway enrichment analysis of the chromatin enrichment data of AzadC-treated mESCs under 2iL conditions compared with the untreated ctrl. using pathfindR. **(N)** Volcano plot of chromatin-enriched proteins of 0.2 $\mu M$ AzadC-treated mESCs in the DNMT-TKO under Lif conditions compared with the untreated control (left side: untreated control; right side: AzadC-treated mESCs in the DNMT-TKO).

the prepared chromatin-associated proteome was as anticipated as an overrepresentation of nuclear- and chromatin-bound proteins in comparison with the total cellular proteome.

Next, we investigated the differences of the chromatin-associated proteome between the untreated and the AzadC-treated wt cells under Lif conditions with a focus on enrichment changes of proteins that are directly involved in DNA repair as assigned by the Reactome pathway knowledge base (R-HSA-73894.5) (38). Upon treatment, we observed an overall massive change of the chromatin-associated proteins. Among the significantly deregulated chromatin-associated proteins, 23 DNA repair–associated proteins were enriched after AzadC treatment and 11 DNA repair–associated proteins were depleted compared with the chromatin fraction of the untreated control (Fig 3H, Supplemental Data 3, Table S1). A highly enriched DNA repair–associated protein was Parp1, which was already reported to be involved in the repair of AzadC-induced DNA lesions via base excision repair (BER) (27) and now confirmed by our complementary approach. Another top enriched hit was the flap endonuclease 1 (Fen1) that belongs to the structure-specific endonucleases (39). Fen1 is important for the maturation of Okazaki fragments during DNA replication, but it is also required to deal with DNA replication stress caused by a stalled replication fork and involved in long-patch BER (40) and (microhomology-mediated) alternative end joining (a-EJ) (41, 42). A-EJ is considered as an alternative NHEJ pathway that results in deletions. Therefore, this error-prone repair pathway is a mutagenic driver, but it is activated when NHEJ and homologous recombination (HR) repair pathways are compromised (43).

Other important repair proteins that we found significantly enriched in the chromatin fraction of AzadC-treated wt cells under Lif conditions were the DNA-dependent protein kinase catalytic subunit Prkdc, the DNA mismatch repair proteins Msh2 and Msh6, the single-strand selective monofunctional uracil DNA glycosylase Smug1, and the DNA repair protein Rad50. Prkdc regulates phosphorylation of H2AX and is one of the key proteins in NHEJ (44, 45). Msh2 and Msh6 are central components of the post-replicative DNA mismatch repair system (MMR) by forming the heterodimer MutS$\alpha$ that binds to DNA mismatches to initiate repair (46). Rad50 on the contrary plays a pivotal role in DSB repair (47). Interestingly, the uracil DNA glycosylase Ung, which belongs to the same protein superfamily as Smug1, significantly depleted from the chromatin fraction of AzadC-treated wt cells (Fig 3H). Smug1 and Ung are both part of the BER machinery to repair the presence of uracil in the DNA after deamination of cytosine. However, although Ung is important for repair of uracil in replicating DNA, Smug1 is more important to remove uracil in non-replicating chromatin (48). Among the depleted proteins on the contrary were the E3 ubiquitin-protein ligase Rad18, which is an essential component to link DNA damage signalling to activation of HR (49), and the DNA repair protein complementing XP-C cell homolog Xpc, which is an integral component of the nucleotide excision repair machinery (NER) (50).

To obtain a holistic picture of the affected cellular pathways, with a focus on the involved DNA repair pathways, we performed a pathway enrichment analysis using pathfindR (51) (Fig 3I, Supplemental Data 3, Table S2). One of the top enriched terms was "Proteasome" with more than eightfold enrichment and 23 contributing proteins being highly enriched in the chromatin fraction of the AzadC-treated wt cells compared with untreated cells (Table 2).

Protein–DNA crosslinks can be repaired by proteasomal activity that digests the bulky protein adducts to the peptide level to allow the canonical repair machinery to excise the affected DNA sites in a next step (52, 53). Our chromatin proteomics data strongly indicated that the proteasome removes the covalently trapped DNMTs, whereas other proteases involved in removal of protein–DNA crosslinks like Spartan could not be detected in the chromatin fraction. As expected, pathway enrichment analysis also revealed that "DNA replication" and "cell cycle" were heavily affected after AzadC treatment as many checkpoint proteins were significantly enriched in the chromatin fraction after AzadC treatment. Among them was the serine/threonine-protein kinase Chek1, which is central for checkpoint-mediated cell-cycle arrest and the activation of the DDR when DNA damage is present (54), as well as all components of the mini-chromosome maintenance (MCM) complex (Mcm2–Mcm7) (Table 2). MCM proteins were reported to play an important role in sensing DNA damage at replication forks and subsequently recruiting the DNA repair machinery (55, 56). Regarding the activated DNA repair pathways, NHEJ, MMR, and BER showed strong enrichment in the pathway analysis in AzadC-treated wt cells compared with untreated cells under Lif conditions (Fig 3I and Table 2). This is in line with the up-regulated chromatin recruitment of key components of those pathways (Fig 3H), indicating that those repair pathways were activated.

To gain better insights into the activation of the DDR in wt cells under Lif conditions, we then analysed the global protein expression-level changes after AzadC treatment (Supplemental Data 3, Table S3) and subsequently compared the chromatin enrichment of the DNA repair–associated proteins with their global expression-level changes (Supplemental Data 3, Table S4). Pathway enrichment analysis of the whole proteome data revealed that BER also showed significant enrichment (Fig 3J) (Supplemental Data 3, Table S5). Among the BER-associated proteins, Smug1 and Fen1 showed both significantly increased expression levels and significantly changed chromatin enrichment (Fig 3K). These results suggested that BER is activated by increased protein expression levels and by active recruitment of the involved DNA repair proteins to the site of action. Strikingly, all other DNA repair–associated proteins that were found to be enriched in the chromatin fraction after AzadC treatment did not show any significant expression-level changes (Fig 3K), revealing that DDR activation in response to AzadC-induced DNA lesions is substantially based on active chromatin recruitment. On the contrary, there were several DNA repair–associated proteins that showed exclusively higher expression levels after AzadC treatment but were not significantly enriched in the chromatin fraction. The most prominent one, with a 16-fold higher protein level after AzadC treatment (–log[$P$-value] = 1.4, log$_2$FC = 4), was the key tumour suppressor p53. P53 is a nuclear transcription factor that plays, among many other functions, a pivotal role in DDR by inducing the expression of many DNA repair–associated genes and genes that trigger cell-cycle arrest and apoptosis in the presence of DNA damage (57). Under normal conditions, p53 has a very short half-life time and is in addition mostly present in an inactivated state. Our whole proteome data are in accordance with literature that upon DNA damage sensing, the p53 protein levels are substantially increased as the *Tp53* mRNA is more efficiently translated and the p53 protein is stabilized,

**Table 2. List of selected significantly enriched pathways after 0.2 μM AzadC treatment over 48 h of wt cells under Lif conditions after having analysed the chromatin-associated proteome.**

| Term | Fold enrichment | *P*-value range | Chromatin-enriched after AzadC treatment | Depleted from chromatin after AzadC treatment |
|---|---|---|---|---|
| Proteasome | 8.79 | $2.15 \times 10^{-27}$–$1.77 \times 10^{-18}$ | Psma1, Psma3, Psma4, Psma7, Psmb2, Psmb3, Psmb6, Psmc1, Psmc2, Psmc3, Psmc4, Psmc5, Psmc6, Psmd1, Psmd2, Psmd3, Psmd7, Psmd8, Psmd11, Psmd12, Psmd13, Psme3 | |
| DNA replication | 6.69 | $6.76 \times 10^{-18}$–$9.59 \times 10^{-7}$ | Fen1, Lig1, Pola1, Pola2, Pold1, Prim1, Rpa2, Mcm2, Mcm3, Mcm4, Mcm5, Mcm6, Mcm7 | Pole3 |
| NHEJ | 3.97 | $2.83 \times 10^{-4}$–$1.60 \times 10^{-2}$ | Prkdc, Rad50, Fen1 | Rrbp1 |
| MMR | 3.74 | $1.57 \times 10^{-7}$–$1.83 \times 10^{-2}$ | Msh2, Msh6, Rpa2, Pold1, Lig1 | |
| BER | 3.65 | $2.59 \times 10^{-8}$–$2.51 \times 10^{-6}$ | Smug1, Lig1, Fen1, Parp1, Pold1 | Ung, Pole3 |
| Cell cycle | 3.11 | $2.83 \times 10^{-14}$–$1.33 \times 10^{-6}$ | Cdk4, Cdkn2a, Ywhaz, Ywhab, Ywhae, Ywhah, Ywhag, Bub1b, Cdc14b, Chek1, Prkdc, Mcm2, Mcm3, Mcm4, Mcm5, Mcm6, Mcm7 | Rbpj, Gtf2b, Cdc20, Mad1l1 |

The proteins that are assigned to the specific pathway according to pathfindR and showed a significant change in chromatin enrichment after treatment are listed.

leading to its accumulation (58). The activation of p53 and the subsequent effects were also observed in the pathway enrichment analysis that showed p53 signalling among the top enriched pathways (Fig 3J, Supplemental Data 3, Table S5). The reason why we did not observe significant enrichment of p53 in the chromatin fraction of AzadC-treated cells might be that transcription factors usually bind very transiently to the DNA and can therefore often not be captured without crosslinking. One of the target genes of p53 is the DNA damage–binding protein 2 (Ddb2), a key component of NER that subsequently forms a complex at the lesion site with the ubiquitin ligase Cullin-4a (Cul4a), which then initiates the proteolytic digest of Ddb2. This targeted degradation at the lesion site in turn recruits the DNA repair protein Xpc to initiate global genome NER (GG-NER) (59, 60). Interestingly, Ddb2 showed a significant increase in global protein levels, but we failed to detect it in the chromatin fraction, and Cul4a and Xpc were even depleted from the chromatin fraction after AzadC treatment (Fig 3H, Supplemental Data 3, Table S1). Having a closer look at other components of the GG-NER, we observed that most of the depleted proteins in the chromatin fraction after AzadC treatment belong to or are at least associated with this repair pathway, including TIMELESS-interacting proteins Tipin and Claspin (Clspn), which are part of the intra-S checkpoint that is activated upon DNA replication stress, the NAD⁺-dependent protein deacetylase sirtuin-6 (Sirt6), and the DNA polymerase e (Pole) (61, 62, 63). Moreover, there were several NER components, including replication factor 1 (Rfc1), general transcription factor IIH subunits Ercc2 and Ercc4 (Xpd and Xpf), and Gtf2H4 and Cyclin-H (Ccnh) (39, 64, 65, 66) that were not enriched in the chromatin fraction after AzadC treatment (Fig 3K), but significantly up-regulated on the global protein expression level. Consequently, pathway enrichment analysis also revealed an up-regulation of NER based on the whole protein expression data (Fig 3J). These combined results suggest that when replication stress is very high after AzadC treatment, NER is initially activated on the

protein expression level but then either not used or even actively excluded from the AzadC-induced DNA lesion repair as it progresses.

As DNMT activity and therefore the induced replication stress upon AzadC treatment are very different in the wt under Lif and 2iL conditions, we replicated the chromatin enrichment workflow in wt cells that were treated with 2.5 μM of AzadC under 2iL conditions, with the respective untreated 2iL wt cells as a control. Under 2iL conditions, we observed 33 DNA repair–associated proteins being enriched in the chromatin fraction after AzadC treatment and only two repair-associated proteins being depleted (Fig S3D, Supplemental Data 3, Table S6). 14 DNA repair–associated proteins were enriched in the chromatin fraction under 2iL and Lif conditions (Fig 3L). Among them were several components of BER and MMR, which were also significantly enriched as pathways under 2iL conditions (Fig 3M, Supplemental Data 3, Table S7). Moreover, Rad50 and Chek1 were highly enriched after AzadC treatment under both conditions, indicating that they play a central role in dealing with AzadC-induced DNA lesions. In addition, we observed after AzadC treatment under 2iL conditions, chromatin enrichment of the Ataxia-telangiectasia mutated (Atm) and Rad3-related (Atr) protein kinases, including their downstream targets Chek1 and Chek2 (Figs 3L and S3C). Atm and Atr are essential for genome integrity, the DDR and checkpoint signalling, and p53 activation (67, 68). p53 signalling was also significantly enriched in the pathway enrichment analysis of the chromatin fraction of AzadC-treated wt cells under 2iL conditions (Fig 3M). Interestingly, the FA repair pathway, including HR, that was previously reported to play an important role in repair of AzadC-induced DNA lesions (26), was under Lif conditions neither enriched as a pathway nor were the DNA repair protein Rad51 homolog 1 (Rad51, also known as Fancr) or other central components of the FA repair pathway enriched in the chromatin fraction of AzadC-treated cells. In contrast, under 2iL conditions, several important FA proteins, including Rad51 and Fanci, were significantly

enriched (Figs 3L and S3D), and HR and FA, but not NHEJ, were overrepresented pathways (Fig 3M). Overall, these results indicated that BER and MMR are commonly activated DDRs to deal with AzadC-induced DNA lesions. In contrast, FA and HR are only activated when DNMT activity and therefore the resulting replication stress after DNMT crosslinking are moderate, whereas NHEJ and a-EJ, which are more error-prone but very efficient, are the repair pathways of choice when replication stress is immense after AzadC treatment because of high DNMT activity. To validate the applicability of our chromatin–proteome-centred approach for comprehensive investigation of DDR after treatment with DNA-damaging reagents in a human AML cell line, we compared the chromatin-enriched DNA repair–associated proteins of AzadC-treated mESCs under 2iL conditions with those of AzadC-treated KG-1 cells. In accordance with existing literature on comparable cancer cell lines (26), proteins of the FA pathway were highly enriched in KG-1 cells, including RAD51 (Fig S3E, Supplemental Data 3, Table S8). Of note, although proteins for HR were enriched in KG-1, several proteins involved in NHEJ were depleted from the chromatin fraction, suggesting that also cancer cells promote, if possible, FA-HR over NHEJ to repair AzadC-induced DNA double-strand breaks.

In the last step, we investigated which repair pathways are involved to deal with genomically incorporated AzadC when DNMT-DNA crosslinking cannot take place. To this end, we treated DNMT-TKO cells under Lif conditions with an equal dose of AzadC compared with wt cells (200 nM over 48 h) and enriched the chromatin protein fraction. The total amount of DNA repair–associated proteins that we could detect in the chromatin fraction of DNMT-TKO cells did not differ from the wt (Supplemental Data 3, Table S9). However, in contrast to wt cells that showed a massive change of the chromatin-bound proteins, hardly any changes of chromatin-associated proteins in general could be observed in the DNMT-TKO cells compared with the untreated control (Fig 3N). Among the few significantly enriched proteins were only three repair-associated proteins: the endonuclease 8-like 3 DNA glycosylase Neil3, the crossover junction endonuclease MUS81, and the WD repeat–containing protein 48 (Wdr48), which is part of the FA repair pathway. To rule out that only the low concentration of AzadC failed to activate DDR in DNMT-TKO cells, we treated the DNMT-TKO cells under 2iL conditions with 2.5 $\mu$M of AzadC over 48 h and checked chromatin enrichment. However, although we observed a loss of cellular fitness of DNMT-TKO cells (Fig 2B) when exposed to this concentration, the chromatin-bound protein fraction was hardly affected and no concerted DDR was observed either (Fig S3F, Supplemental Data 4, Table S10). These results indicated that without DNMT crosslinking, AzadC does not invoke a strong and concerted DDR that can be monitored on the bulk chromatin level.

### The cytotoxic effect of AzaC depends on its DNA incorporation and DNMT activity but not on crosslinking of the RNA methyltransferase Trdmt1

To elucidate the RNA-dependent effect of AzaC in the absence of DNMTs, we compared treatment of wt and DNMT-TKO cells under 2iL and Lif conditions with increasing concentrations of AzaC, because an RNA effect should manifest in both genotypes and would be potentially dominant in the AzaC-treated DNMT-TKO mESCs (Fig 1G,

Table 1). First, we checked the incorporation levels of AzaC into DNA as AzadC (Fig 4A) and into RNA as AzaC (Fig 4B) after 2.5 $\mu$M AzaC feeding for 48 h under 2iL conditions. For the wt, we observed the expected lower incorporation rate into DNA with ~12.5% AzadC/G intensity compared to feeding with AzadC after 24 h (Fig 4A, wt treatment a). This result is in perfect accordance with literature where it was reported that 10–20% of AzaC are incorporated into DNA (12). The pattern that the DNA-incorporated levels dropped by half 48 h after treatment when no additional compound was added, but could be increased by additional 80% when a new compound was added after 24 h, was also observed for AzaC treatment in the wt. However, although we did not detect any difference regarding the uptake and metabolization of AzadC between wt and DNMT-TKO cells (Fig 2F), AzaC incorporation into DNA was significantly lower in the DNMT-TKO compared with the wt (Fig 4A). On the contrary, the incorporation rates of AzaC into RNA were comparable between the wt and the DNMT-TKO cells (Fig 4B). Therefore, different intercellular metabolization kinetics and not a difference in the uptake of AzaC were probably the reason for the lower DNA incorporation in the DNMT-TKO. Moreover, the level of AzaC in RNA in wt and DNMT-TKO mESCs could only be maintained but not further increased when an additional compound was added after 24 h (Fig 4B). This observation indicated that RNA compared with DNA has a higher turnover and AzaC could therefore not accumulate. Next, we investigated the phenotypic changes after AzaC treatment (2.5 $\mu$M for 48 h, treatment b) in wt and DNMT-TKO cells under 2iL conditions. Whereas the proliferation rate initially slowed down in the first 24 h after AzaC treatment in the DNMT-TKO cells, proliferation quickly resumed in those cells and the cell number after 48 h was the same as for the untreated control (Fig 4C). In contrast, the wt showed a persistent slower proliferation rate after AzaC treatment with doubling times comparable to AzadC treatment. Intriguingly, the slower proliferation rate in the wt was not reflected in higher cytotoxicity as indicated by brightfield microscopy (Fig 4D) and the flow cytometry–based apoptosis assay (Fig 4E). In the DNMT-TKO, AzaC did not have any cytotoxic activity in the applied concentration either (Fig 4D and E). When we had a closer look at the DNA hypomethylating effect of AzaC compared with AzadC in the wt cells under 2iL conditions, we observed a substantial decrease in mdC for both compounds with no difference between AzadC and AzaC at 1.25 and 2.5 $\mu$M (Fig 4F). In contrast, γH2AX levels in the wt were much higher after AzadC compared with AzaC treatment as indicated by the intensity of the γH2AX signal in fluorescence microscopy (Fig 4G). In leukaemic cells, it was reported that AzaC incorporation into RNA destabilizes the mRNA of *RRM2*, which results in lower RRM2 protein expression levels. As the ribonucleotide reductase is essential to produce DNA nucleotides, depletion of one of its core subunits ultimately leads to a proliferation block (13). To check whether this mechanism also contributes to the inhibition of proliferation in AzaC-treated mESCs, we checked our 2iL whole proteome data of AzaC-treated wt and DNMT-TKO cells for proteins involved in nucleotide metabolism according to Reactome (R-HAS-15869.7) but found only the glutathione reductase significantly down-regulated in the wt (Supplemental Data 3, Table S11), and thymidine kinase (Tk) as well as inosine-5′-monophosphate dehydrogenase 1 (Impdh) and adenine phosphoribosyltransferase (Aprt) down-regulated in the DNMT-TKO (Supplemental Data 4, Table

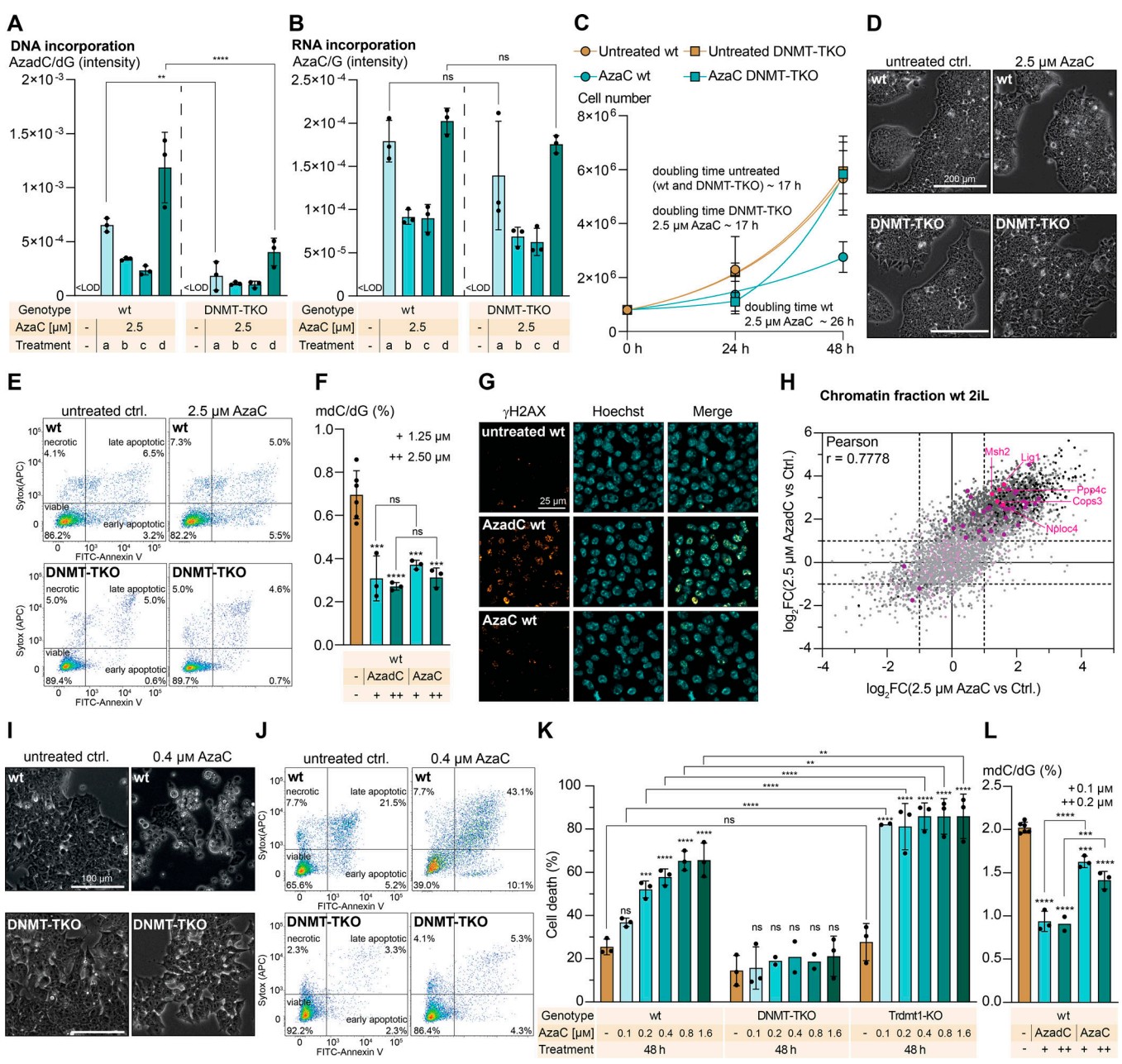

**Figure 4.  Effect of AzaC treatment on proliferation rate and viability in mESCs.**
**(A, B)** Intensity of AzadC signal normalized to the intensity of dG signal in genomic DNA (A) and of AzaC in RNA (B), measured by QQQ-MS, in the wt and the DNMT-TKO after treatment with 2.5 μM of AzaC under 2iL conditions. LOD = limit of detection; treatment a = AzaC addition at 0 h, harvest after 24 h; b = AzaC addition at 0 h, harvest after 48 h; c = AzaC addition at 0 h, medium change after 24 h to medium without AzaC, harvest after 48 h; d = AzaC addition at 0 h, medium change after 24 h to medium with freshly added AzaC, harvest after 48 h. The bar represents the mean, error bars represent the SD, and each dot represents one biologically independent replicate. Two-way ANOVA (genotype and treatment) combined with Šídák's multiple comparisons test to compare the same treatment between the two genotypes (Supplemental Data 1, Fig 4A and B). **(C)** Proliferation curve of wt and DNMT-TKO cells after treatment with 2.5 μM of AzaC under 2iL conditions compared with the untreated controls, which are also displayed in Fig 2C. For each sample to be measured, 800,000 cells were seeded initially (0 h). For the 24-h and the 48-h timepoints, three biologically independent replicates were quantified. The symbol represents the mean, and the error bar represents the SD. Fitting of the growth curve for wt (untreated and treated) and untreated DNMT-TKO by exponential (Malthusian) growth with the constraint $Y_0 = 800{,}000$. (Supplemental Data 1, Fig 4C). **(D)** Representative brightfield microscopy images of wt and DNMT-TKO cells untreated or after treatment with 2.5 μM AzaC under 2iL conditions for 48 h. **(E)** Representative flow cytometry scatter plots of wt and DNMT-TKO cells untreated and after 48-h treatment with 2.5 μM AzaC under 2iL conditions (n = 10,000 events per condition) using FITC–Annexin V binding as a marker for apoptosis and SYTOX Red as a marker for dead cells. **(F)** Amount of mdC, quantified by QQQ-MS and normalized to the amount of dG, in the wt under 2iL conditions after 48-h treatment with AzadC or AzaC compared with the untreated control. Ctrl. and AzadC data are also displayed in Fig 2A. The bar represents the mean, error bars represent the SD (SD), and each dot represents one biologically independent replicate. One-way ANOVA combined with Tukey's multiple comparisons test (Supplemental Data 1, Fig 4F). Stars above bars of AzadC- or AzaC-treated samples indicate a significant difference in the mean compared with the untreated control. **(G)** Fluorescence microscopy images of γH2AX signal in wt cells (2iL conditions) untreated or after 48-h treatment with 5.0 μM AzadC or AzaC. Hoechst staining shows nuclei. **(H)** Correlation plot of the chromatin-enriched proteome changes after Aza treatment (x-axis) and AzadC treatment (y-axis) compared with the untreated control in wt cells under 2iL conditions. Repair-associated

S12). When we checked the chromatin-enriched proteome of wt cells after AzaC treatment (Supplemental Data 4, Table S13), we observed a very high correlation compared with the AzadC-treated cells (Pearson's r = 0.7778; Supplemental Data 4, Table S14), but only few repair-associated proteins reached the significance threshold (Fig 4H), which is in line with the observation that AzaC introduces less DNA damage than AzadC. In summary, the obtained results from the DNMT-TKO suggest that under steady conditions, the RNA incorporation of AzaC leads to an initial proliferation delay that can be, however, quickly overcome by the cells because of high RNA turnover. As a consequence, no persistent RNA-dependent effect can be achieved without additional AzaC supply. As DNA damage seemed to be marginal under 2iL conditions after AzaC treatment in the wt, which is in line with the absent toxicity, the inhibition of DNMTs and the resulting significant decrease in mdC appear to be responsible for the significantly decreased proliferation rate in the wt after AzaC treatment. Although AzaC incorporation into gDNA reaches only 12.5% compared with AzadC treatment in the wt, the changes of the chromatin-associated proteins are highly similar for AzaC and AzadC treatment but less pronounced after AzaC treatment.

Next, we investigated the effect of AzaC when the transcriptome and proteome and therefore cellular identity of both genotypes drastically change by switching the mESC culture conditions from naïve to primed. In stark contrast to 2iL conditions, AzaC treatment of wt cells under Lif conditions had a very strong effect on the cellular viability as indicated by brightfield microscopy (Fig 4I) and the flow cytometry–based apoptosis assay (Fig 4J). Similar to AzadC treatment, cellular proliferation of the wt completely stopped using an AzaC concentration as low as 0.2 $\mu$M (Fig S4A). The effect on the viability of DNMT-TKO cells, however, was minimal at this concentration (Fig 4I and J), and the proliferation rate was not affected either (Fig S4A). To further investigate the RNA-dependent features of AzaC, we quantified cell death in wt, in DNMT-TKO, and in addition in Trdmt1-deficient (Dnmt2-deficient) mESCs (Trdmt1-KO) after exposing the cells to increasing AzaC concentrations for 48 h under Lif conditions. In the Trdmt1-KO, no Trdmt1 crosslinking to RNA by AzaC can take place. For the wt, we observed a steady increase in toxicity with increasing concentrations of AzaC, whereas for the DNMT-TKO, AzaC was not toxic at the applied concentrations (Fig 4K). Remarkably, the Trdmt1-KO was significantly the most sensitive genotype towards AzaC treatment. Based on these data, we could not explain why the Trdmt1-KO reacted so sensitively towards AzaC treatment, but this observation contradicts the idea that Trdmt1-RNA crosslinking as a result of RNA incorporation of AzaC has a

negative impact on cellular well-being, at least in the chosen mESC model system.

Last, we wanted to check whether DNA damage or decreased mdC levels were responsible for the substantially increased cell death events in the wt under Lif conditions. First, we quantified mdC levels in the untreated control, AzadC-, and AzaC-treated cells and observed that although mdC levels were significantly decreased after AzaC treatment compared with the untreated control, the levels remained at significantly higher levels compared with AzadC-treated cells with no difference between the two concentrations tested (Fig 4L). Under 2iL conditions, mdC levels were equally reduced after AzaC and AzadC treatment (Fig 4F); cytotoxicity of AzaC, however, remained low. In contrast, under Lif conditions, AzaC was significantly less efficient compared with AzadC to reduce mdC (Fig 4L), but cytotoxicity was high. Therefore, we concluded that the induction of substantial DNA damage after AzaC treatment because of high DNMT activity was the underlying reason for the dramatically increased sensitivity of the primed wt mESCs towards AzaC compared with the naïve wt mESCs. To substantiate this hypothesis, we in addition tested the non-nucleoside DNMT inhibitor RG-108 (32) on wt cells under Lif conditions, and although we reached a reduction in mdC comparable to the reduction after AzaC treatment (Fig S4B), we failed to detect increased cell death (Fig S4C), showing that the reduced mdC levels did not impair cellular viability under these conditions.

In summary, the effects of AzaC appear to primarily rely on its incorporation into DNA, and although DNMT inhibition and thereby reduced mdC levels significantly slow down the proliferation rate, the induction of cell death depends on the severity of the induced DNA damage. At moderate DNMT activity, the level of severe DNA lesions and therefore cell death rates remain low after AzaC treatment, whereas under high DNMT activity, AzaC has substantial cytotoxicity comparable to AzadC.

## Discussion

Our comparison of wt mESCs with DNMT-TKO mESCs revealed that the presence of AzadC in the genome contributes to its toxicity profile independent from DNA-DNMT crosslinking. However, although its presence results in DNA lesions with increasing concentrations, a concerted DDR that can be monitored on the whole chromatin level in a bulk experiment is only invoked after DNA-DNMT crosslinking. In addition, we could show that at high DNMT activity, the formation of DNA-DNMT crosslinks and the resulting

proteins that were significantly enriched (–log[*P*-value] > 1.3 and |log$_2$FC| >1) after both treatments are labelled magenta dots. Repair-associated proteins that were only significant after AzadC treatment are in dark purple, and non-significant repair–associated proteins are displayed in light purple. Pearson's correlation is shown. **(I)** Representative brightfield microscopy images of wt and DNMT-TKO cells untreated or after treatment with 0.4 $\mu$M AzaC under Lif conditions for 48 h. **(J)** Representative flow cytometry scatter plots of wt and DNMT-TKO cells untreated and after 48-h treatment with 0.4 $\mu$M AzaC under Lif conditions (n = 10,000 events per condition) using FITC–Annexin V binding as a marker for apoptosis and SYTOX Red as a marker for dead cells. **(K)** Summary of cell death events (necrotic + early apoptotic + late apoptotic) in wt, DNMT-TKO, and Trdmt1-KO cells (Lif conditions) after 48-h treatment with AzaC in increasing concentrations compared with the untreated control. Bars represent the mean, error bars represent the SD, and dots represent biologically independent replicates. Two-way ANOVA (genotype and treatment) with Šídák's multiple comparisons test to compare the same treatment between the three genotypes and to compare with the control within one genotype (Supplemental Data 1, Fig 4K). Stars above bars of AzaC-treated samples indicate a significant difference in the mean compared with the respective untreated control. **(L)** Amount of mdC, quantified by QQQ-MS and normalized to the amount of dG, in the wt under Lif conditions after 48-h treatment with AzadC or AzaC compared with the untreated control. The bar represents the mean, error bars represent the SD (SD), and each dot represents one biologically independent replicate. One-way ANOVA combined with Tukey's multiple comparisons test (Supplemental Data 1, Fig 4L). Stars above bars of AzadC- or AzaC-treated samples indicate a significant difference in the mean compared with the untreated control. **(A, B, F, K, L)** ns $P_{adj}$ > 0.05, * 0.05 > $P_{adj}$ > 0.01, ** 0.01 > $P_{adj}$ > 0.001, *** 0.001 > $P_{adj}$ > 0.0001, **** $P_{adj}$ < 0.0001.

DNA damage are the dominating MoA of AzadC and AzaC, even at low concentrations, whereas an RNA-dependent effect could not be observed in our model system. Nevertheless, AzaC shows better results for some haematological cancers in the clinic than AzadC (13, 69). One reason can be different uptake and metabolization kinetics of AzaC compared with AzadC, but also tumour-specific dependencies and RNA, protein, and nucleotide metabolism, which are more complex in the clinical context and not reflected by our model. DNA demethylation by DNMT inhibition, on the contrary, seemed to have an impact on the proliferation rate but not on the viability of the cells. Furthermore, although the presence of AzadC in the genome results in DNA-DNMT crosslinks, which trigger a strong DDR, the global removal of AzadC from the genome primarily depends on passive dilution by ongoing replication. To keep the level of genomically incorporated AzadC high, a constant exposure to AzadC has to be guaranteed as it is expected from the low stability of AzadC towards hydrolysis (70) and now showed by our time course experiments following different treatment regimens.

In this study, we used mESCs as a model because they feature an intact DDR, which was important to investigate the DNA repair mechanisms in a comprehensive way. Moreover, they allow control over DNMT activity and dynamics by changing the culturing conditions, thereby mimicking cancer types with a very different DNMT activity profile without having a different genetic background. Using chromatin enrichment followed by proteome analysis of the enriched fraction, we systematically analysed the involved repair mechanisms to deal with the AzadC-induced DNMT-DNA crosslinks under different levels of DNMT activity. Under moderate and high DNMT activity, the DNA-DNMT crosslinks are targeted by the proteasome and trigger MMR and BER. Using our approach, we confirmed the previously reported involvement of PARP1 (27) and FA-dependent HR (26) to repair AzadC-induced lesions. Furthermore, our results indicate that under high DNMT activity, with consequently very high replication stress after DNA-DNMT crosslinking, FA-dependent repair does not contribute anymore to the repair of AzadC-induced DNA lesions but is replaced by NHEJ and a-EJ. In the next step, the here acquired information on the MoAs of AzadC and AzaC has to be transferred to clinically relevant models and our initial study with KG-1 cells confirms that the workflow is applicable. We have shown here that the DDR towards the repair of AzadC-induced DNA lesions depends on multiple parameters and cannot be investigated by whole transcriptome and proteome studies. Importantly, depending on the severity of damage, cells can adapt their DDR to deal with AzadC-induced lesions as there is not one specific DNA repair pathway that is required under any circumstances. This observation suggests that cancer cells can acquire resistance against AzadC by modifying their DDR and that inhibition of a few specific DNA repair proteins like PARP1 might lead to initially promising synergistic effects but can be overcome quickly.

Overall, the here presented method to compare proteome information after chromatin enrichment with whole proteome data revealed highly relevant proteome changes for chromatin dynamic-relevant processes that would have remained undiscovered if only whole proteome changes or transcriptome changes would have been investigated. These results emphasize that the spatial resolution of the cellular proteome is equally important as temporal resolution of protein expression changes over time to study the cellular response towards any chromatin-targeting treatment.

# Materials and Methods

If not indicated otherwise, Milli-Q grade water was used for all experiments and room temperature (RT) refers to a temperature between 20°C and 22°C.

## Reagents and cell lines

### Chemicals
All chemicals that were used in this study are listed in Table 3. If not indicated otherwise, chemicals were used without further purification and stored according to the available product sheet.

AzadC and AzaC were dissolved in pre-cooled water to a concentration of 10 mM, directly shock-frozen in liquid nitrogen, and stored in 10 $\mu$l aliquots at –80°C. Aliquots were only thawed once before addition and diluted to a 100 $\mu$M solution with water. The integrity of AzadC and AzaC stocks was routinely checked by HPLC, followed by MS of the peak fraction.

RG108 was dissolved in DMSO to a concentration of 100 mM, directly shock-frozen in liquid nitrogen, and stored in 2.5 $\mu$l aliquots at –80°C. Aliquots were only thawed once before addition and directly diluted to 10 mM using 10% (vol/vol) EtOH in water.

### Cell culture media and supplements
For culturing mESCs, the medium components as listed in Table 4 were used.

Media and supplements were stored according to the available product sheet. Supplements that were shipped in dry form were solubilized and stored appropriately before use. $\beta$-Mercaptoethanol was diluted to a concentration of 50 mM in PBS and stabilized with 35 $\mu$M EDTA (pH 8.0). mLif was diluted to $10^6$ U/ml with PBS containing 15% (wt/vol) BSA, sterile-filtered, aliquoted, and stored at 4°C for up to two months. FBS was used without heat inactivation.

The mESC basic medium consisted of DMEM, 10% (vol/vol) FBS, 2 mM L-alanyl-L-glutamine, 0.1 mM $\beta$-mercaptoethanol, 1x MEM-NEAA, and 1x Pen-Strep. After preparation, the basic medium was sterile-filtered using a bottle top filter unit ($\emptyset$ 0.2 $\mu$m). To prepare Lif medium from basic medium, mLif was added to a final concentration of $10^3$ U/ml. To prepare 2iL medium from basic medium, mLif was added to a final concentration of $10^3$ U/ml, and CHIR99021 and PD334581 were added to a final concentration of 3 $\mu$M. Basic and Lif media were stored for up to 2 wk at 4°C, and 2iL medium was stored for up to 10 d at 4°C.

### Antibodies
All antibodies used in this study are listed in Table 5.

## Biological resources

wt J1 mESCs were described in Li et al (1992) (71) and originally provided by the Jaenisch laboratory (Whitehead Institute, USA). DNMT-TKO J1 mESCs were described in Tsumara et al (2006) (29) and originally provided by the Okano laboratory (RIKEN). Trdmt1-KO J1 mESCs were described in Okano et al (1998) (72) and originally provided by the Jaenisch lab.

**Table 3. List of chemicals used in this study.**

| Name | Manufacturer | CAS number | Catalogue number |
|---|---|---|---|
| Acetonitrile, LC-MS grade (MeCN) | Roth | 75-05-8 | AE70.2 |
| Ammonium bicarbonate (ABC) | Sigma-Aldrich | 1066-33-7 | 09830 |
| Annexin V Binding Buffer | BioLegend | / | 422201 |
| 5-Aza-2'-deoxycytidine | Biosynth | 2353-33-5 | ID74843 |
| 5-Azacytidine | Biosynth | 320-67-2 | NA02947 |
| BCA protein assay kit | Thermo Fisher Scientific | / | 23227 |
| Benzonase nuclease (25.3 units/µl) | Millipore | / | 70746-3 |
| Bisbenzimide H33342 (Hoechst) | Sigma-Aldrich | 23491-52-3 | 382065 |
| Bradford reagent | Bio-Rad | / | 5000006 |
| Bromophenol blue | Sigma-Aldrich | 115-39-9 | 114391 |
| ChemiBLOCKER | Millipore | / | 2170 |
| 1,4-Dithiothreitol (DTT) | Sigma-Aldrich | 3483-12-3 | D0632 |
| DMSO | Sigma-Aldrich | 67-68-5 | D8418 |
| Dynabeads Protein G | Invitrogen | / | 100004D |
| EDTA | Sigma-Aldrich | 60-00-4 | E9884 |
| EGTA | Sigma-Aldrich | 67-42-5 | E3889 |
| Ethanol abs. (EtOH) | Sigma-Aldrich | 67-42-5 | E3889 |
| FITC–Annexin V | BioLegend | / | 640906 |
| Fluoroshield mounting medium | Sigma-Aldrich | / | MKCP4984 |
| 16% formaldehyde solution (w/v) | Thermo Fisher Scientific | / | 28906 |
| Formic acid, LC-MS grade (FA) | Thermo Fisher Scientific | 800-874-3723 | 85178 |
| Gelatine from porcine skin | Sigma-Aldrich | 9000-70-8 | G2500 |
| Glycerol | Sigma-Aldrich | 56-81-5 | G5516 |
| Glycine | Sigma-Aldrich | 56-40-6 | G8898 |
| Guanidine thiocyanate | Sigma-Aldrich | 593-84-0 | G9277 |
| Hepes | Sigma-Aldrich | 7365-45-9 | H3375 |
| Hydrochloric acid, 37% (HCl) | Sigma-Aldrich | 7647-01-0 | 320331 |
| Iodoacetamide (IAA) | Sigma-Aldrich | 144-48-9 | I1149 |
| Methanol (MeOH) | Thermo Fisher Scientific | 67-56-1 | M/4056/17 |
| Nucleotide Digestion Mix | New England Biolabs | / | M0649S |
| NP-40 | Sigma-Aldrich | 9016-45-9 | 74385 |
| Dulbecco's phosphate-buffered saline without $MgCl_2$/$CaCl_2$ (PBS) | Sigma-Aldrich | / | D8537 |
| Dulbecco's phosphate-buffered saline with $MgCl_2$/$CaCl_2$ (PBS+) | Sigma-Aldrich | / | D8662 |
| Phosphatase inhibitor cocktail 2 | Sigma-Aldrich | / | P5726 |
| Phosphatase inhibitor cocktail 3 | Sigma-Aldrich | / | P0044 |
| Ponceau S solution | Sigma-Aldrich | 6226-79-5 | P7170 |
| Potassium hydroxide (KOH) | Sigma-Aldrich | 1310-58-3 | 221473 |
| Protease inhibitor (cOmplete EDTA free) | Roche Diagnostics | / | 43203100 |
| RG108 | Sigma-Aldrich | 48208-26-0 | 24724594 |
| RNase A | Millipore | / | 70856 |
| Skim milk powder | Millipore | / | 70166 |

**Table 3.  Continued**

| Name | Manufacturer | CAS number | Catalogue number |
|------|--------------|------------|------------------|
| Sodium chloride (NaCl) | Sigma-Aldrich | 7647-14-15 | S5886 |
| Sodium dodecyl sulphate (SDS) | Sigma-Aldrich | 151-21-3 | L6026 |
| Sodium hydroxide (NaOH) | Sigma-Aldrich | 1310-73-2 | S8045 |
| SuperSignal West Pico Chemiluminescent Substrate | Thermo Fisher Scientific | / | 34077 |
| SYTOX Red Dead Cell Dye | Life Technologies Corporation | / | S34859 |
| Trizma base (Tris) | Sigma-Aldrich | 77-86-1 | T1503 |
| Trisodium citrate | Sigma-Aldrich | 6132-04-3 | S4641-500G |
| Triton X-100 | Sigma-Aldrich | 9036-19-5 | T8787 |
| Trypan Blue Stain 0.4° | Thermo Fisher Scientific | / | T10282 |
| TrypLE Express | Life Technologies Corporation | / | 12604-013 |
| Trypsin, LC-MS grade (0.5 µg/µl) | Thermo Fisher Scientific | / | 90305 |
| Tween-20 | Sigma-Aldrich | 9005-64-5 | P9416 |
| Water, LC-MS grade ($H_2O$) | Honeywell | 7732-18-5 | 39253 |

**Table 4.  List of medium components for mESC culturing.**

| Medium component | Manufacturer | CAS number | Catalogue number |
|------------------|--------------|------------|------------------|
| β-Mercaptoethanol | Sigma-Aldrich | 60-24-2 | F63689 |
| CHIR99021 | Axon Medchem | 252917-06-9 | HY-10182 |
| DMEM—high glucose | Sigma-Aldrich | / | D6429 |
| MEM Non-essential Amino Acid (NEAA) Solution (100x) | Sigma-Aldrich | / | M71145 |
| ESGRO Recombinant Mouse LIF Protein (mLif) | Sigma-Aldrich | / | ESG1107 |
| L-Alanyl-L-glutamine | Sigma-Aldrich | 39537-23-0 | G8541 |
| Pansera ES-grade FBS | Pan Biotech | / | P30-2602 |
| PD334581 | Axon Medchem | 391210-10-9 | HY-10254 |
| Penicillin–streptomycin (Pen-Strep) (100x) | Sigma-Aldrich | / | AP0781 |

**Table 5.  List of antibodies used in this study. mAB = monoclonal antibody; pAB = polyclonal antibody.**

| Antibody | Manufacturer | Catalogue number | Application |
|----------|--------------|------------------|-------------|
| Anti-phospho-histone H2A.X (Ser139), clone JBW301 (mouse mAB) | Millipore | 05-636 | Western blotting (1:1,000), Immunofluorescence (1:250) |
| Anti-histone H3, clone D1H2 (rabbit mAB) | Cell Signaling | 4970S | Western blotting (1:1,500) |
| Anti-PARP1 (rabbit pAB) | Proteintech | 13371-1-AP | Western blotting (1:1,000) |
| Alexa Fluor 488–conjugated AffiniPure Rat Anti-Mouse IgG (H+L) | Jackson ImmunoResearch | 415-545-166 | Immunofluorescence (1:400) |
| HRP-conjugated AffiniPure Goat Anti-Mouse IgG (H+L) | Proteintech | SA00001-1 | Western blotting (1:7,500–1:10,000) |
| HRP-conjugated AffiniPure Goat Anti-Rabbit IgG (H+L) | Proteintech | SA00001-2 | Western blotting (1:7,500–1:10,000) |

## Statistical analysis

For the proteomics experiments, details about the statistical analysis, including sample size, data exclusion, and significance thresholds, are given in the Proteomics—Materials and Methods section, including the respective analysis software. For all other experiments, GraphPad Prism (v 9.4.0) was used for statistical analysis and details are given in Supplemental Data 1. No statistical methods were used to pre-determine the sample size. Sample sizes were chosen based on cost, experience, and commonly used sample sizes for in vitro experiments (n ≥ 3), which provided in this study low intersample variety between samples of the same group.

The number of samples for each experiment is either described in the Materials and Methods section or directly evident from the respective figure and figure legend.

### Data availability

Original (unprocessed) and metadata are deposited as described in the Data Availability section.

### Database references

We used the Reactome knowledge database (https://reactome.org, accessed March 2023–April 2024 (38)) to assign DNA repair–relevant proteins in our proteomics data sets. To receive optimal coverage, we mapped the mouse proteins to the respective human proteins via the gene name and did the subsequent analysis on the human pathways.

The UniProt database (https://uniprot.org) was used to download the FASTA file of *Mus musculus*.

GOrilla (https://cbl-gorilla.cs.technion.ac.il/; accessed February 2023 (73)) was used for Gene Ontology (GO) term analysis.

### mESC handling

#### Culture conditions and passaging

mESCs were cultivated at 37°C in water-saturated, $CO_2$-enriched (5%) atmosphere on gelatine-coated plates. For gelatine coating, 0.2% (wt/vol) gelatine in water was prepared, heat-sterilized, brought to RT, and filtered. Afterwards, culture dishes were coated for 10–60 min at 37°C, coating solution was aspirated, and mESCs were directly plated in the appropriate amount of medium. For mESC maintenance, 2iL medium was used as a standard medium. mESCs were routinely passaged every 2–3 d in a ratio of 1:4 to 1:8 when reaching a confluency of 60–75%. To detach the mESCs, medium was aspirated, cells were washed with PBS, and TrypLE (150 $\mu$l/six well) was added for 4–5 min at 37°C before trypsinisation was stopped with medium. Then, mESCs were resuspended to a single-cell solution, and the required amount of cells was centrifuged at 300$g$ for 3 min at RT and afterwards replated in new medium. When reaching passage #25 after thawing, cells were discarded. mESCs were checked once during cultivation for Mycoplasma contamination using a PCR-based Mycoplasma detection kit (#PP-401L; Jena Bioscience) as indicated by the manufacturer.

#### Priming

For priming, the anticipated portion of mESCs were cultured after passaging in Lif medium instead of 2iL. After 48 h, mESCs on Lif medium were passaged again in new Lif medium. If not indicated otherwise, cells were primed for 96 h in total before analysis.

#### Treatment

Experiments were only started if cell and colony morphology indicated high cellular fitness and cell viability was > 90% as indicated by trypan blue staining. In all experiments, the "Untreated Ctrl." refers to mESCs of the same genotype for a respective experiment, which was not treated with any compound, but the same

way otherwise. For each experiment, untreated and treated samples were seeded from the same mESC batch and then handled in parallel. The number of seeded cells depended on plate size: ca. 115,000 cells for a 12 well, ca. 300,000 cells for a six well, ca. 700,000 cells for a p60, ca. 1,900,000 cells for a p100, and ca. 4,500,000 cells for a p150. AzadC and AzaC were added at the indicated concentrations from the diluted 100-$\mu$M stock solution directly into the medium after seeding. Unless stated otherwise, medium was not changed anymore until cell harvest 48 h after the start of the treatment. For treatment under naïve conditions, 2iL medium was used (2iL, 48-h treatment). For treatment under primed conditions, mESCs were primed for 48 h before treatment and the compounds were added to Lif medium after the second passaging in Lif medium (in total 96 h primed in Lif, treatment in the last 48 h). For RG108 experiments, mESCs were maintained for 4 wk in 2iL medium containing 50 $\mu$M of RG108. For priming, mESCs were seeded for 48 h in Lif medium containing 50 $\mu$M of RG108 before they were finally seeded into Lif medium containing 200 $\mu$M of RG108 for an additional 48 h.

#### Proliferation assay

For the proliferation assay, two wells were seeded per biologically independent sample type. After 24 h, the first well per sample was harvested and cells were counted, and after 48 h, the second well per sample was harvested and the cells were counted. Counting was done using a Countess 3 automated cell counter (Invitrogen).

### Microscopy

#### Brightfield microscopy

For brightfield microscopy images, cells were imaged directly in the medium at the end of treatment using an EVOS M5000 imaging system in transmission mode and 10x or 20x magnification. Afterwards, brightness and contrast were automatically adjusted using Adobe Photoshop 2023 for optimal visualization of the cells. Pictures were taken from representative regions.

#### Fluorescence confocal microscopy

All steps were performed in a humidity chamber and at RT if not otherwise specified. 30,000 cells per well were seeded in ibidi $\mu$-slide 8 Well (#80826; ibidi) and treated as indicated. After 48-h treatment, cells were washed with PBS+ and fixed for 10 min using 4% formaldehyde solution. After three times of washing with PBS+, the cells were permeabilized and blocked for 30 min using 0.3% (vol/vol) Triton X-100 and 5% (vol/vol) ChemiBLOCKER. The primary anti-γH2A.X antibody was diluted in PBS+, containing 5% (vol/vol) CB and 0.3% (vol/vol) Triton X-100, and applied overnight at 4°C. After incubation, mESCs were washed three times with PBS+ containing 2% (vol/vol) CB. For secondary detection, the fluorescent-labelled Alexa Fluor 488 anti-mouse antibody was diluted in PBS+, containing 3% (vol/vol) CB, and applied for 1 h in the dark, followed by three times of washing with PBS+. Cell nuclei were stained with Hoechst 33342 (5 $\mu$g/ml), which was applied for 15 min in the dark, followed by one washing step with PBS+. After mounting, the samples were analysed using a Leica SP8 confocal laser scanning microscope with associated LAS X software (Leica). Regions for imaging were chosen based on the Hoechst signal. Brightness and

contrast were adjusted for the control using ImageJ (version 1.54a), and afterwards, the settings were applied to all other images.

## Triple-quadrupole mass spectrometry (QQQ-MS) for nucleoside quantification

For QQQ-MS experiments, mESCs were seeded in a p60 and treated with the indicated concentrations. For the AzadC and AzaC incorporation experiments, cells were treated for either 24 h (treatment a), 48 h without medium change (treatment b), 48 h with medium change after 24 h without second compound addition (treatment c) or 48 h with medium change after 24 h and second compound addition at the same concentration (treatment d). For all other QQQ-MS experiments, mESCs were treated for 48 h without medium change (treatment b). After treatment was finished, medium was aspirated, and mESCs were washed with PBS and harvested. After harvest, mESCs were washed once with PBS and afterwards directly lysed in 800 $\mu$l of GTC buffer (3.5 M guanidine thiocyanate, 25 mM trisodium dihydrate, 14.3 mM $\beta$-mercaptoethanol, pH 6.9), and either processed directly or shock-frozen in liquid $N_2$ and stored at −80°C. For thawing, lysed samples were quickly warmed to RT and gDNA isolation was performed.

### gDNA and RNA isolation
After cell lysis, gDNA and RNA isolation was performed according to Traube et al (2019) (34) with minor modifications. Butylated hydroxytoluene and deferoxamine were not added to the washing buffers, and mESCs were only lysed by adding the chaotropic GTC buffer, but the bead mill step described in the previously published protocol was skipped. For AzadC and AzaC incorporation experiments, isolated gDNA or RNA was directly subjected to nucleoside digest and QQQ-MS measurements. For mdC quantification, the isolated gDNA could be stored at −80°C before nucleotide digestion.

### Nucleotide digestion
Nucleotide digestion was performed in technical duplicates per biologically independent sample. Two digestion controls were added for each digestion as described in Traube et al (2019) (34). Per sample, 0.5 $\mu$g of gDNA or RNA was digested in a total volume of 30 $\mu$l using 0.5 $\mu$l of enzyme and 3 $\mu$l of 10x buffer from the Nucleotide Digestion Mix for 1 h at 37°C. Afterwards, 20 $\mu$l of water was added to reach a final volume of 50 $\mu$l. For AzadC and AzaC incorporation experiments, samples were filtered immediately as described in Traube et al (2019) (34). After filtration, the samples were directly subjected to QQQ-MS. For mdC quantification, the nucleoside mixture could be stored at −20°C before filtering and analysing.

### QQQ-MS data acquisition
For QQQ-MS, an Agilent 1290 Infinity equipped with a variable wavelength detector (VWD) combined with an Agilent Technologies G6490 Triple Quad LC/MS system with electrospray ionization (ESI-MS; Agilent Jetstream) was used. All solvents were LC-MS grade. The operating parameters were as follows: positive-ion mode, cell accelerator voltage of 5 V, $N_2$ gas temperature of 120°C and $N_2$ gas flow of 11 Litre/min, sheath gas ($N_2$) temperature of 280°C with a flow of 11 Litre/min, capillary voltage of 3,000 V, nozzle voltage of 0 V, nebulizer at 60 psi, high-pressure RF at 100 V, and low-pressure RF at 60 V. The instrument was operated in a dynamic MRM mode (Supplemental Data 4, Tables S15, S16, and S17). For separation, a Poroshell 120 SB-C8 column (2.7 $\mu$m, 2.1 × 150 mm; #683775-906; Agilent Technologies) was used. Running conditions were 35°C and a flow rate of 0.35 ml/min for all experiments. Specifications for AzadC incorporation experiments were (Supplemental Data 4, Table S15) as follows: binary mobile phase of 5 mM $NH_4OAc$ aqueous buffer A (pH 5.3) and an organic buffer B of 0.0075% FA in MeCN. The gradient started at 100% solvent A for 1.5 min, followed by an increase of solvent B to 20% over 7 min (1.5–8.5 min) and further to 80% B within the following minute (8.5 −9.5 min). 80% B was maintained for 2.5 min (9.5 −12 min) before returning to 100% solvent A in 0.5 min and a 2.2-min re-equilibration period. Specification of AzaC incorporations was (Supplemental Data 4, Table S16) as follows: 0.0075% FA in aqueous buffer A and an organic buffer B of 0.0075% FA in MeCN. The gradient started at 100% solvent A for 1.2 min, followed by an increase of solvent B to 5% over 5 min (1.2 −6.2 min) and further to 80% B within the following 1.3 min (6.2 −7.5 min). 80% B was maintained for 2 min (7.5 −9.5 min) before returning to 100% solvent A in 0.5 min and a 2.5-min re-equilibration period. Specification of mdC quantification was (Supplemental Data 4, Table S17) as follows: binary mobile phase of 0.0075% FA in aqueous buffer A and an organic buffer B of 0.0075% FA in MeCN. The gradient started at 100% solvent A, followed by an increase of solvent B to 3.5% over 4 min (0 −4 min), and from 4 to 7 min, solvent B was further increased to 5%. From 7.0 to 8.0 min, solvent B was increased to 80% and maintained at 80% for 2.5 min before returning to 100% solvent A in 1.5 min and a 2.2-min re-equilibration period. Of each sample, 10 $\mu$l was co-injected with 1 $\mu$l of stable isotope-labelled internal standard (ISTD). The ISTD mix consisted for all measurements of 200 $\mu$M theophylline to have an ISTD mix-UV control, and of 0.5 $\mu$M of each isotope standard ($^{15}N_5$-$^{13}C_{10}$-dA, $^{13}C_9$-dC, $^{15}N_5$-$^{13}C_{10}$-dG, $^{15}N_2$-$^{13}C_5$-dT, $D_3$-m$^5$dC, $^{15}N_2$-$D_2$-hm$^5$dC, $^{15}N_2$-f$^5$dC, $^{15}N_2$-ca$^5$dC, and $^{15}N_5$-8oxodG). Calibration curves for canonical nucleosides (dA, dC, dG, and dT) spanned 0.1 −200 pmol and for the modified nucleosides (mdC, hmdC, fdC, cadC, and 8oxodG) 0.004 −5 pmol.

### Analysis
The sample data were analysed by quantitative and qualitative MassHunter software from Agilent (v B07.01). As there was no internal standard available for exact quantification of AzadC and AzaC, we calculated the area under the curve ratio AzadC/dG for DNA measurements, or for RNA measurements, the AzaC/G ratio to obtain the relative incorporation levels normalized to the dG and G content. For mdC quantification, we followed the procedure as described in Traube et al (2019) (34), except that also the amount of canonical nucleosides was calculated by MS and not via the UV trace. Samples where the sum of C-modifications (dC + mdC + hmdC) deviated by more than 15% from dG were discarded as they did not pass the quality threshold. As we measured each sample in technical duplicates, we calculated the mean of the technical replicates to obtain the mean for each biologically independent sample.

## Flow cytometry–based apoptosis assay

For flow cytometry, mESCs were seeded in a 12 well and treated as indicated. Before harvest, the medium, which includes dead floating cells, was not aspirated but collected as well and the remaining attached cells were harvested and combined with the cells from the medium. Afterwards, mESCs were washed twice with PBS and subsequently counted. $1.5 \times 10^5$ cells per sample were transferred into a new tube. Apoptosis and necrosis were determined using FITC–Annexin V Apoptosis Detection Kit and SYTOX Red Dead Cell Stain. To this end, cells were resuspended in 150 $\mu$l of Annexin V binding buffer supplemented with 0.75 $\mu$l of FITC-conjugated Annexin V and 0.15 $\mu$l of SYTOX Red Dead Cell Stain, gently vortexed, and incubated at RT for 15 min in the dark. Afterwards, samples were put on ice and the cell suspension was filtered through a 35-$\mu$m strainer before measurement. For the analysis, BD FACSCanto (recording of 10,000 events per sample) and FlowJo Single Cell Analysis Software (v10.8.0) were used. Gates (FSC [A] – SSC [A] to remove cell debris, FSC [A] – FSC [H] to gate for single cells, and last FITC/APC to distinguish between live, dead, and early apoptotic cells) were set once for the control sample and then applied to all other samples.

## Immunoblot analysis

### Nuclear extract preparation

For the preparation of nuclear extracts, which were used for Western blotting to enrich nuclear-specific proteins, mESCs were seeded in a p100 and treated for 48 h. After treatment, the mESCs were harvested and nuclear extracts were prepared as previously described by Dignam et al (1983) (74) with the modification that every buffer was supplemented with phosphatase inhibitor cocktail 2 and phosphatase inhibitor cocktail 3, 1:100 each. Furthermore, cOmplete protease inhibitor was used to inhibit any protease activities. Afterwards, the protein concentration was determined using a Bradford assay as described by the manufacturer. SDS loading buffer (final concentration 50 mM Tris–HCl [pH 6.8], 100 mM DTT, 2% [wt/vol] SDS, 10% [vol/vol] glycerol, 0.25% [wt/vol] bromophenol blue) was added. The samples were vortexed, incubated for 5 min at 92°C, and afterwards stored at –20°C. Before loading the samples on a polyacrylamide gel, the samples were heated for an additional 2 min at 92°C and vortexed thoroughly.

### Western blotting (Immunoblot)

Per sample, 15 $\mu$g of nuclear extract in SDS loading buffer was loaded on a 4–15% precast polyacrylamide gel (#4561083EDU; Bio-Rad) and Color-coded Prestained Protein Marker, Broad Range (10–250 kD) (#P7719S; New England Biolabs) was used as a protein standard. The gel was run at constant 150 V for 60 min in SDS running buffer (25 mM Tris, 192 mM glycine, 0.1% [wt/vol] SDS). For blotting, we used a PVDF blotting membrane (Amersham Hybond P0.45 PVDG membrane #10600023; GE Healthcare) and pre-cooled Towbin blotting buffer (25 mM Tris, 192 mM glycine, 20% [vol/vol] MeOH, 0.038% [wt/vol] SDS). The membrane was activated for 1 min in methanol, washed with water, and equilibrated for an additional 2 min in Towbin blotting

buffer; the Whatman gel blotting papers (#WHA10426981; Sigma-Aldrich) were equilibrated for 15 min in Towbin buffer, and the precast gel was equilibrated for 5 min in Towbin buffer after the run. Western blotting (tank [wet] electrotransfer) was performed at 4°C for 9 h at constant 35 V. After blotting, the PVDF membrane was blocked for 1 h at RT and constant shaking using 5% (wt/vol) skim milk powder in TBS-T (20 mM Tris–HCl [pH 7.5], 150 mM NaCl, 0.1% [vol/vol] Tween-20). The primary antibodies were diluted in 5 ml of 5% (wt/vol) skim milk powder in TBS-T. The blocking suspension was discarded, and the diluted primary antibodies were added for 12–16 h at 4°C and shaking. After incubation, the primary antibodies were discarded, and the membrane was washed three times for 10 min with TBS-T. HRP-conjugated secondary antibodies were diluted in 5% (wt/vol) milk powder in TBS-T and added for 1 h at room temperature under shaking. Afterwards, the membrane was washed two times with TBS-T and one time with TBS (TBS-T without Tween-20) before SuperSignal West Pico Chemiluminescent Substrate was used for imaging. Western blots were imaged using Amersham Imager 680 (auto exposure mode).

For imaging the same blot multiple times using different antibodies, the membrane was directly stripped after imaging. To this end, the membrane was put in TBS-T and the buffer was heated in a microwave until boiling. Afterwards, the buffer was discarded and the procedure was repeated in total three times. After stripping, the membrane was blocked again using 5% (wt/vol) milk powder in TBS-T and the protocol followed the above-described procedure.

At the end, the membrane was stained with Ponceau S to visualize the total protein load.

## Liquid chromatography–tandem mass spectrometry (LC-MS/MS)—proteomics

For proteomics experiments, mESCs were seeded in a p150 and treated for 48 h. For each sample type (specific combination of genotype, culturing conditions, treatment), four biologically independent replicates were initially generated. After treatment, mESCs were harvested and washed with PBS. 10% of the cells were used for whole proteome isolation and the rest for chromatin enrichment to generate whole proteome and chromatin-enriched proteome samples from the same batch of cells.

### Whole proteome isolation

Cells were lysed in 200 $\mu$l of total lysis buffer (20 mM Hepes, 1% [vol/vol] NP-40, and 0.2% [wt/vol] SDS) for 30 min on ice and afterwards centrifuged at 21,000$g$ at 4°C for 10 min. The supernatant containing the proteins was transferred to a new tube.

### Chromatin enrichment

Chromatin extraction was performed according to Gillotin (2018) (75). The cell pellet was lysed in 150 $\mu$l (ca. 5x pellet size) of E1 lysis buffer containing 50 mM Hepes–KOH, 140 mM NaCl, 1 mM EDTA, 10% (vol/vol) glycerol, 0.5% (vol/vol) NP-40, 0.25% (vol/vol) Triton X-100, 1 mM DTT, and cOmplete protease inhibitor. The cell extract was then centrifuged at 1,100$g$, 4°C for 2 min, and the supernatant

(cytoplasmic fraction) was transferred into a new tube. The pellet was resuspended again in E1 lysis buffer, incubated for 10 min on ice, and then centrifuged again at 1,100$g$, 4°C for 2 min. The supernatant was discarded. Next, the pellet was washed three times in 50 $\mu$l ice-cold E2 buffer containing 1 M Tris–HCl (pH 8.0), 200 mM NaCl, 1 mM EDTA, 0.5 mM EGTA, and cOmplete protease inhibitor, by resuspending the pellet fully and centrifuging at 1,100$g$, 4°C for 2 min. The supernatants (nuclear fraction) were transferred and pooled in a new tube. In the last of the three washing steps, the sample was incubated on ice for 10 min before centrifugation. After centrifugation, the remaining pellet was resuspended in E3 buffer containing 1 M Tris–HCl (pH 7.5), 20 mM NaCl, 1 mM MgCl$_2$, 0.1% (vol/vol) Benzonase, and cOmplete protease inhibitor. The pellets were resuspended in this buffer, and sonicated for 10 cycles with 30 " on and 30 " off at maximum power at 4°C using a Bioruptor Pico sonication device (Diagenode). Afterwards, the samples were centrifuged at 16,000$g$, 4°C for 10 min. The supernatant containing the chromatin-bound proteins was transferred into a new tube for further analysis.

Protein concentrations were determined by the bicinchoninic acid assay using the bicinchoninic acid assay protein assay kit according to the manufacturer's protocol. Every sample was measured in technical duplicates at 562 nm on a multimode microplate reader (Tecan), and the mean of the two technical replicates was calculated for each sample. The total protein concentration was calculated using a calibration curve prepared with BSA that was re-done for every measurement. Because of poor quality and low protein amount, one sample of the whole proteome fraction for each sample (2iL and Lif) could not be analysed and had to be discarded, leaving three biologically independent replicates per sample type for whole proteome measurement.

### SP3 protocol

20 $\mu$g of protein was used for each sample. The protein sample was added to pre-washed Dynabeads Protein G (bead/protein ratio 10:1, ca. 7 $\mu$l of resuspended beads for 200 $\mu$g) and filled up with total lysis buffer to a working volume of 50 $\mu$l. The samples were incubated on the beads while shaking in an Eppendorf ThermoMixer C at 1,000 rpm for 1 min at RT. 120 $\mu$l of EtOH was added, and then, the samples were incubated again for 5 min while shaking at 1,000 rpm. The beads were trapped on a magnet for 2 min, and the supernatant was discarded. The bead-bound proteins were then washed three times with 100 $\mu$l 80% (vol/vol) ethanol, and incubated for 1 min while shaking at 850 rpm for each step, and then, the supernatant was discarded on a magnet. The beads were resuspended in 100 $\mu$l of 100 mM ABC buffer, then 10 mM DTT and 20 mM IAA were added from a 1-M stock solution, respectively, and the samples were incubated for 5 min at 95°C while shaking at 850 rpm. After the samples were cooled down to RT, trypsin protease (LC-MS grade) was added to the sample at a ratio of 1:50 (0.4 $\mu$g of trypsin for 20 $\mu$g of protein) and incubated overnight at 37°C while shaking at 850 rpm. The peptides were then carefully transferred into a fresh tube, and the beads were washed twice with 50 $\mu$l 0.1% (vol/vol) FA. The three fractions were pooled, again incubated on a magnet, and transferred into a fresh tube to remove all beads from the sample. Then, the peptide samples could be stored at –80°C until further analysis.

### MS acquisition and analysis

MS analysis was performed as described in Makarov et al (2022) (76) on an Orbitrap Eclipse Tribrid mass spectrometer (Thermo Fisher Scientific) coupled to an UltiMate 3000 Nano HPLC (Thermo Fisher Scientific) via an EASY-Spray source (Thermo Fisher Scientific) and FAIMS interface (Thermo Fisher Scientific) using a data-independent acquisition (DIA) mode. LC-MS grade solvents were used. Per sample, 1 $\mu$g of peptides was first loaded on Acclaim PepMap 100 $\mu$-precolumn cartridge (5 $\mu$m, 100 A, 300 $\mu$m ID × 5 mm; Thermo Fisher Scientific) and was then separated at 40°C on a PicoTip emitter (non-coated, 15 cm, 75 $\mu$m ID, 8 $\mu$m tip; New Objective) that was *in-house*–packed with the Reprosil-Pur 120 C18-AQ material (1.9 $\mu$m, 150 Å, Dr. A. Maisch GmbH). A gradient over 60 min and 0.1% (vol/vol) FA in LC-MS grade water as buffer A and 0.1% (vol/vol) FA in MeCN as buffer B were used with a flow rate of 0.3 $\mu$l/min and 0–5 min 4% B, then from 5 to 6 min to 7% B, followed by 6 –36 min to 24.8% B, 36 –41 min to 35.2% B. From 41 to 41.1 min, B was increased to 80% until 46 min, when column was re-equilibrated at 4% B until 55 min. The DIA duty cycle consisted of one MS1 scan followed by 30 MS2 scans with an isolation window of the 4 m/z range, overlapping with an adjacent window at the 2 m/z range. MS1 scan was conducted with Orbitrap at 60,000 resolution power and a scan range of 200–1,800 m/z with an adjusted RF lens at 30%. MS2 scans were conducted with Orbitrap at 30,000 resolution power, and RF lens was set to 30%. The precursor mass window was restricted to a 500–740 m/z range. HCD fragmentation was enabled as activation type with a fixed collision energy of 35%. FAIMS was performed with one CV at –45 V for both MS1 and MS2 scans during the duty cycle.

### Analysis of MS spectra and pathway analysis

MS raw files were processed in DIA-NN (v. 1.8.1) (77) to determine the protein identity and quantity in each sample. FASTA digest for library-free search/library generation was activated, and a FASTA spectral library generated for *mus musculus* from UniProt (uniprot.org) was used. As protease, trypsin/P was chosen, the missed cleavages being allowed were set to 2, and the minimal peptide length was set to 6. MBR (match between runs) mode was enabled. As quantification strategy, Robust LC (high precision) was used and threads were set to 7. Otherwise, default settings were used. From the DIA-NN output, the protein group file was uploaded to and subsequently analysed using MaxQuant–Perseus (v. 1.6.15) (78). First, samples where the number of identified unique proteins (1% FDR) was <500 were discarded to avoid analysis bias by many missing values that would have to be imputed in the further analysis steps. Next, samples were grouped according to the genotype, culturing condition, and treatment (samples that are biologically independent but of the same type). Proteins had to be present in at least >50% of the samples within at least one sample group or were otherwise completely discarded from the analysis to avoid bias by a large number of missing values over all experiments. For the remaining proteins, LFQ intensities were log$_2$-transformed and missing values were replaced from normal distribution (separately for each column, default settings, Perseus). The wt and the DNMT-TKO samples were separately analysed using the volcano plot function in Perseus (*t* test, both-sided, 250 randomizations, FDR 0.05), and treated samples were analysed against the respective controls. We applied a stricter significance threshold for the chromatin-

enriched proteins with a significance threshold of $-\log(P\text{-value}) > 1.3$ and $|\log_2\text{FC}| > 1$ (|fold change| > 2) compared with the whole proteome where a significance threshold of $-\log(P\text{-value}) > 1.3$ and $|\log_2\text{FC}| > 0.58496$ (|fold change| > 1.5) was applied, because more preparation steps were required to isolate the chromatin-bound proteins. This can increase intersample variation independent from biological reasons. Furthermore, we wanted to ensure to take only proteins into account for the downstream analysis that showed a very strong chromatin recruitment as a response to AzadC treatment. $\text{Log}_2\text{FC}$ refers to "Difference" in Perseus.

Next, DNA repair proteins were assigned according to Reactome (38) projecting the mouse proteins to the human equivalents. For identifying enriched pathways within the proteomics data sets, the pathfindR tool was used as described (51). Pathways were considered as significantly enriched when the criteria fold enrichment $\geq 2$ and highest $P$-value $< 0.05$ were both fulfilled. GO analysis was performed using GOrilla (73) to analyse the chromatin fraction against the whole proteome of untreated wt cells with the settings specified in Supplemental Data 3.

### KG-1 handling

KG-1 cells were cultivated at 37°C in water-saturated, $CO_2$-enriched (5%) atmosphere in RPMI-1640 medium, supplemented with 10% FBS and 2 mM L-alanyl-L-glutamine, and maintained at a concentration of $0.5 \times 10^6$ cells/ml. For AzadC treatment, $1.0 \times 10^6$ cells/ml were plated in 5 ml and treated for 72 h with 1.25 $\mu$M AzadC. Untreated cells served as a control. Four biologically independent replicates per condition were used for subsequent nuclear enrichment and MS analysis.

### Supplementary Data

Supplementary Data are provided in different files as indicated in the main text. Information on the statistical analysis of different data is provided in Supplemental Data 1. Information on the GO-term analysis when chromatin-associated proteome enrichment was compared with whole proteome data can be found in Supplemental Data 2. All other information regarding the different proteomics data sets is provided in Supplemental Data 3. Supplementary data that are relevant to the Materials and Methods section are provided in Supplemental Data 4. All Supplementary Data files are available online.

## Data Availability

The mass spectrometry proteomics data have been deposited to the ProteomeXchange Consortium (79) via the PRIDE (80) partner repository with the data set identifier PXD045353.

Other original data, analysis files, and respective metadata, which are not in the supplementary files, have been deposited using figshare.com and can be accessed via the following:

Doi: 10.6084/m9.figshare.24146604 (apoptosis assay data).
Doi: 10.6084/m9.figshare.24146592 (QQQ data).

## Supplementary Information

## Acknowledgements

We thank Dr. Matthias Heiß and Kerstin Kurz for QQQ maintenance, troubleshooting, and quality checks, and Dr. Matthias Heiß also for his valuable input for the QQQ measurements. We thank Dr. Pavel Kielkowski, Dmytro Makarov, and Andreas Wiest for Eclipse maintenance, calibration, and troubleshooting. We thank the Biomedical Center Munich Core Facility Flow Cytometry (LMU) for their support. We thank Johannes Pforr (TUM Natural School of Sciences) for his help with initial experiments for this study and Corinna Pleintinger for checking the integrity of AzaC and AzadC at the HPLC. FRT and SM thank the Deutsche Forschungsgemeinschaft (DFG) for financial support via CRC1309 (Grant Nr. 325871075, Projects B05 [S Michalakis] and C08 [FR Traube]). FR Traube thanks the Daimler und Benz Stiftung (Grant Nr. 32-09/21), the Fonds der Chemischen Industrie (Liebig Fellowship), and the TUM Junior Fellows Fond for support. M Däther thanks the Fond der Chemischen Industrie for PhD Fellowship. Funding for Open Access Charge: DFG via CRC1309.

### Author Contributions

T Aumer: data curation, formal analysis, supervision, validation, investigation, and writing—review and editing.
M Däther: data curation, formal analysis, supervision, validation, investigation, and writing—review and editing.
L Bergmayr: data curation, formal analysis, supervision, and investigation.
S Kartika: validation, investigation, and writing—review and editing.
T Zeng: investigation.
Q Ge: investigation.
G Giorgio: investigation.
AJ Hess: investigation.
S Michalakis: formal analysis, supervision, funding acquisition, and writing—review and editing.
FR Traube: conceptualization, data curation, formal analysis, supervision, funding acquisition, validation, investigation, visualization, methodology, project administration, and writing—original draft.

### Conflict of Interest Statement

The authors declare that they have no conflict of interest.

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
