## [Reviewer comments · Life Science Alliance]

Life Science Alliance

The type of DNA damage response after Decitabine treatment depends on the level of DNMT activity

Tina Aumer, Maïke Däther, Linda Bergmayr, Stephanie Kartika, Theodor Zeng, Qingyi Ge, Grazia Giorgio, Alexander Hess, Stylianos Michalakis, and Franziska Traube

DOI: <https://doi.org/10.26508/lsa.202302437>

Corresponding author(s): Franziska Traube, University of Stuttgart

Review Timeline:

Submission Date:	2023-10-15
Editorial Decision:	2023-11-27
Revision Received:	2024-04-24
Editorial Decision:	2024-05-29
Revision Received:	2024-06-08
Accepted:	2024-06-11

Transaction Report:

November 27, 2023

Re: Life Science Alliance manuscript #LSA-2023-02437-T

Prof. Franziska R Traube
University of Stuttgart
Institute of Biochemistry and Technical Biochemistry
GERMANY

Dear Dr. Traube,

Thank you for submitting your manuscript entitled "The type of DNA damage response after Decitabine treatment depends on the level of DNMT activity" to Life Science Alliance. The manuscript was assessed by expert reviewers, whose comments are appended to this letter. We invite you to submit a revised manuscript addressing the Reviewer comments.

Thank you for this interesting contribution to Life Science Alliance. We are looking forward to receiving your revised manuscript.

Sincerely,

B. MANUSCRIPT ORGANIZATION AND FORMATTING:

Reviewer #1 (Comments to the Authors (Required)):

Aumer et al. investigate how the DNMT inhibitors decitabine and azacytidine affect DNA damage. The authors show that decitabine induces a DNMT-dependent DNA damage response that depends on the severity of decitabine-induced DNA lesions. While this is an interesting manuscript, I have several comments that I hope could improve the study:

- The authors mention that they used a "spatial proteomics" approach to study azacytidine and decitabine. Can the authors clarify what is meant by spatial proteomics? The LC-MS/MS experiments appear to be performed either on whole cells or on the chromatin fraction of bulk cells.
- The study mentions that mESCs are a good model system to study the mode of action of both drugs. However, the results in Figure 1C/D suggest that there are substantial differences between WT and DNMT1 KO cells after priming compared to the naive state. Some additional analysis may be warranted here to show the biological processes/pathways that are unique to each genotype/condition.
- Figure 1G is somewhat confusing (at least to this reviewer). How is this panel generated? Is this presented as a summary of the proteomics datasets? Some clarification would be important for readers.
- There should also be a proliferation curve or cell cycle flow for all concentrations of AzadC shown in Fig. 2B.
- The RNA-dependent effects of 5-aza have also been shown to be due to inhibition of ribonucleotide reductase subunits. Its RNA-based effects may also be more complicated than observed in the manuscript. A discussion would be important to place the study findings in the context of the available literature.

Reviewer #2 (Comments to the Authors (Required)):

In this manuscript, Tina et al. aim to address the contribution of individual MoA of AzadC and AzaC respectively. They utilized the wild-type ES and DNMT TKO ES systems to distinguish the contributions of DNA-DNMT crosslink, DNA damage, and RNA-TRDMT1 crosslink in the cell toxicity induced by AzadC and AzaC under three different conditions: DNMT non-expression, low expression, and high expression. The authors showed that when the expression levels of DNMT are low in cells, AzadC-induced proliferation inhibition and cell apoptosis mainly occur through DNA damage induction that is not related to DNMT-DNA crosslink. When DNMT expression is high, DNA-DNMT crosslink, rather than DNA hypomethylation or RNA-related processes, is the main cause of AzadC cell toxicity, and the sensitivity of cells to AzadC is significantly higher than that of cells with low DNMT expression. Similarly, the cell toxicity of AzaC also depends mainly on its level of incorporation into DNA and on the activity of DNMT, while RNA-TRDMT1 crosslink has little contribution to the cell toxicity, at least in ES cells. The authors also identified DNA repair pathways potentially responsive to the DNA lesions induced by AzadC in a DNMT-dependent manner. While the conclusions outlined in this manuscript appear to lack strong novelty, they are generally substantiated by the data. Nevertheless, there is room for improvement in the current manuscript.

Major concerns

1. ES cells have a robust DNA damage repair system, but tumor cells do not. Therefore, using ES cells to study DNA damage response caused by AzadC cannot accurately reflect the choices of tumor cells. This should be taken into consideration and discussed appropriately.
2. In Figure 1C, the majority of the altered proteins are different between the two cell lines, suggesting significant differences between wt and TKO in the process of differentiation. For the commonly changed genes, the authors should list which genes they are, whether they are key genes, and provide evidence for the "comparable" cellular changes.
3. In Figure 2, it appears that the extent of DNA damage caused by AzadC is different between wt and DTKO. However, the amount of AzadC incorporated into the genome is similar, and the proportion of induced cell apoptosis is also similar. How would one reconcile the inconsistency between different levels of DNA damage and similar phenotypes?
4. It should be considered to evaluate the expression of genes involved in regulating AzadC metabolism, such as cytidine deaminase and deoxycytidine kinase, in prime vs naïve, wt vs DNMT TKO, to see if there are any changes that potentially affect the cytotoxicity of AzadC.

5. The authors did not explain why primed DNMT TKO ES cells are more sensitive to AzadC compared to naive ones. This phenomenon does not necessarily support the conclusion that "the DNMT-dependent effect is highly dominant for the toxicity profile of AzadC when DNMT activity is high and DNMT-independent DNA lesions only play a subordinate role".
6. The third section of the Results contains excessive background information unrelated to the result descriptions, which leads to a lack of coherence. It includes unnecessary details, such as the introduction of a specific protein's function. Please revise and shorten this section.
7. The authors observed that under 2iL conditions, there were no changes in chromatin-bound proteins and no activation of DDR in DNMT TKO cells. How does AzadC kill these cells? Please discuss.
8. The authors observed a significantly lower incorporation of AzadC into the DNA of DNMT TKO cells compared to that of the wildtype cells when treated with AzaC. What could be the potential reasons for this difference?
9. It is intriguing that Trdmt1 KO cells are the most sensitive genotype to AzaC treatment, even independent of dosage (Fig 4K). This should be discussed.

Minor concerns

1. What is the "non-canonical nucleobase 5-aza-cytosine"? What MoA does it correspond to?
2. In Figure 2b, there is no statistical data on the survival of cells. How is it demonstrated that there is a difference in cell viability between high and low concentrations? From the figure, the impact of various concentrations on cells appears to be similar for both wildtype (starting at 1.25) and TKO (starting at 2.5) cells.
3. In Figure 2G, H2AX should be used as the loading control to quantify rH2AX. Gray analysis should be presented. The internal reference, H3, was overexposed and had too many nonspecific bands.
4. Figure 2I shows that 1.25um has a similar cytotoxic effect on TKO and wt (wildtype) cells, which is contradictory to Figure 2B that shows that 1.25um does not have a significant cytotoxic effect on TKO cells. Please clarify.

Point-to-point reply

We thank both the reviewers for their valuable comments and suggestions.

Main changes compared to the previous version include:

- In the new manuscript, we tried to be more concise and shortened the manuscript by taking out redundant parts.
- We included now a preliminary study in KG-1 (leukemia cell line) to test the application of the chromatin-centered proteomics workflow for investigating the DNA damage response after AzadC treatment in a relevant model.
- We checked in the whole proteome data whether we see differences in nucleotide metabolism after AzadC and AzaC treatment, but could not find substantial changes. As AzaC works better for some tumor subtypes than AzadC, but this is not reflected by our mESC model system, we formulated the conclusions from the AzaC findings more carefully.

Reviewer #1 (Comments to the Authors (Required)):

Aumer et al. investigate how the DNMT inhibitors decitabine and azacytidine affect DNA damage. The authors show that decitabine induces a DNMT-dependent DNA damage response that depends on the severity of decitabine-induced DNA lesions. While this is an interesting manuscript, I have several comments that I hope could improve the study:

- The authors mention that they used a "spatial proteomics" approach to study azacytidine and decitabine. Can the authors clarify what is meant by spatial proteomics? The LC-MS/MS experiments appear to be performed either on whole cells or on the chromatin fraction of bulk cells.

We are very sorry that the term "spatial" in the abstract was misleading. We wanted to emphasize that we did not only investigate whole proteome changes in this study, but had a specific look at proteome changes in chromatin, which is the site of action of hypomethylating agents. We clarified that now in the revised version and replaced the term "spatial proteomics" in the abstract.

- The study mentions that mESCs are a good model system to study the mode of action of both drugs. However, the results in Figure 1C/D suggest that there are substantial differences between WT and DNMT1 KO cells after priming compared to the naive state. Some additional analysis may be warranted here to show the biological processes/pathways that are unique to each genotype/condition.

We agree that there are substantial cellular differences between the WT and the DNMT-TKO during the transition from the naïve to the primed state. This is not surprising, giving that DNA methylation is one of the major upregulated processes between the naïve and primed state in the wt. We didn't go into details about the differences in the manuscript as the focus is not on mESC development and the role of DNMTs there. In this study, the relevant similarity for us was the fact both genotypes undergo changes upon priming, and even though they are different on the proteome level, the extent (how many proteins are up/down) and the phenotypic consequences (faster proliferation and change of morphology) are very comparable, which implies that there are major changes in transcriptional and translational processes for both of them. To avoid confusion, we have rephrased that part.

- Figure 1G is somewhat confusing (at least to this reviewer). How is this panel generated? Is

this presented as a summary of the proteomics datasets? Some clarification would be important for readers.

Figure 1 G displays the different modes of action from Table 1 and how much each mode of action is presumably contributing to the efficacy profile. The aim of Figure 1 G was to illustrate the different modes of action and was meant as a visual help to understand the underlying rationale of this study. Since it failed its goal and was apparently rather confusing, we have removed it now.

- There should also be a proliferation curve or cell cycle flow for all concentrations of AzadC shown in Fig. 2B.

We have included now the proliferation assay data for the other concentrations in the supplement and the calculated doubling times are in line with our brightfield microscopy images that suggest slower proliferation at higher concentrations for both genotypes.

Doubling times:	wt Ctrl	17 – 19 h
	wt 1.25 μ M AzadC	24 h
	wt 2.5 μ M AzadC	26 h
	wt 5.0 μ M AzadC	63 h
	DNMT-TKO Ctrl.	17 – 18 h
	DNMT-TKO 1.25 μ M AzadC	20 h
	DNMT-TKO 2.5 μ M AzadC	26 h
	DNMT-TKO 5.0 μ M AzadC	43 h

- The RNA-dependent effects of 5-aza have also been shown to be due to inhibition of ribonucleotide reductase subunits. Its RNA-based effects may also be more complicated than observed in the manuscript. A discussion would be important to place the study findings in the context of the available literature.

We thank the reviewer very much for pointing out that we have missed this important mechanism of AzaC. However, in the mESCs (WT and TKO) we cannot observe a significant reduction of Rrm2 expression levels under 2iL and Lif conditions. For uridine monophosphate synthetase, which was also reported to be affected by AzaC, we did not find differential expression levels either. We have included in the revised manuscript a short section in the context of the existing literature and discuss that in our system, at least with the concentration we had applied, we didn't see an effect of AzaC on ribonucleotide metabolism.

Reviewer #2 (Comments to the Authors (Required)):

In this manuscript, Tina et al. aim to address the contribution of individual MoA of AzadC and AzaC respectively. They utilized the wild-type ES and DNMT TKO ES systems to distinguish the contributions of DNA-DNMT crosslink, DNA damage, and RNA-TRDMT1 crosslink in the cell toxicity induced by AzadC and AzaC under three different conditions: DNMT non-expression, low expression, and high expression. The authors showed that when the expression levels of DNMT are low in cells, AzadC-induced proliferation inhibition and cell apoptosis mainly occur through DNA damage induction that is not related to DNMT-DNA crosslink. When DNMT expression is high, DNA-DNMT crosslink, rather than DNA hypomethylation or RNA-related processes, is the main cause of AzadC cell toxicity, and the sensitivity of cells to AzadC is significantly higher than that of cells with low DNMT expression. Similarly, the cell toxicity of

AzaC also depends mainly on its level of incorporation into DNA and on the activity of DNMT, while RNA-TRDMT1 crosslink has little contribution to the cell toxicity, at least in ES cells. The authors also identified DNA repair pathways potentially responsive to the DNA lesions induced by AzadC in a DNMT-dependent manner. While the conclusions outlined in this manuscript appear to lack strong novelty, they are generally substantiated by the data. Nevertheless, there is room for improvement in the current manuscript.

Major concerns

1. ES cells have a robust DNA damage repair system, but tumor cells do not. Therefore, using ES cells to study DNA damage response caused by AzadC cannot accurately reflect the choices of tumor cells. This should be taken into consideration and discussed appropriately.

We thank the Reviewer for pointing this out. It was already part of the discussion, but we have now clarified this part. To test the activation of DDR in a more comprehensive way, we used the mESC model system because it has a robust DDR and by changing the cultivation conditions, DNMT activity can be changed as well. One of the key findings of this study is that there does not seem to be one specific DNA repair pathway that is activated to repair AzadC-induced lesions in any case. We found Fanconi anemia repair and homologous recombination highly activated under moderate or steady-state DNMT activity (which reflects the situation in most tumor cells), but this can be replaced by NHEJ if necessary. Nucleotide excision repair was upregulated on the whole proteome level but its components were then excluded from the chromatin fraction in our model system. However, this doesn't have to be the case in other cells types. As a consequence, a possible escape mechanism of tumor cells is to adjust to AzadC-treatment could be to adapt the DNA repair mechanisms as well. We rephrased that part in the discussion.

2. In Figure 1C, the majority of the altered proteins are different between the two cell lines, suggesting significant differences between wt and TKO in the process of differentiation. For the commonly changed genes, the authors should list which genes they are, whether they are key genes, and provide evidence for the "comparable" cellular changes.

We agree that there are substantial cellular differences between the WT and the DNMT-TKO during the transition from the naïve to the primed state. This is not surprising, giving that DNA methylation is one of the major upregulated processes between the naïve and primed state in the wt. We didn't go into details about the differences in the manuscript as the focus is not on mESC development and the role of DNMTs there. In this study, the relevant similarity for us was the fact both genotypes undergo changes upon priming, and even though they are different on the proteome level, the extent (how many proteins are up/down) and the phenotypic consequences (faster proliferation and change of morphology) are very comparable, which implies that there are major changes in transcriptional and translational processes for both of them. To avoid confusion, we have rephrased that part.

3. In Figure 2, it appears that the extent of DNA damage caused by AzadC is different between wt and DTKO. However, the amount of AzadC incorporated into the genome is similar, and the proportion of induced cell apoptosis is also similar. How would one reconcile the inconsistency between different levels of DNA damage and similar phenotypes?

In the wildtype, AzadC in the genome leads to AzadC-DNMT crosslinks that induce in any case a severe DNA damage. In the DNMT-TKO DNA damage from genomically incorporated AzadC only arises when AzadC undergoes hydrolyzation to form an abasic site. Therefore, it is expected that the extent of DNA damage is higher in the wt. That the phenotypic outcome is not dramatically different can be explained by the fact that under 2iL conditions, DNMT expression

is low and therefore, the DNA damage is still “moderate” compared to Lif conditions, where DNMT expression is high and therefore also the phenotypic outcome is very different between WT and DNMT-TKO.

4. It should be considered to evaluate the expression of genes involved in regulating AzadC metabolism, such as cytidine deaminase and deoxycytidine kinase, in prime vs naïve, wt vs DNMT TKO, to see if there are any changes that potentially affect the cytotoxicity of AzadC.

We thank the reviewer for pointing this out and we checked that now. However, we did not observe strong differences in nucleotide metabolism in our whole proteome data and therefore we have not discussed it (only mentioned it now in the context of AzaC).

For checking, we have exported the list of participating proteins in nucleotide metabolism from reactome and checked the expression level changes of those genes in the whole proteome data in wt and DNMT-TKO after AzadC treatment under 2iL and Lif conditions. Under Lif conditions, more proteins of the nucleotide metabolism were significantly affected in wt and DNMT-TKO, but not in a way that can explain the very high toxicity of AzadC in the WT compared to 2iL and compared to the DNMT-TKO.

Summary:

WT – 2iL (AzadC vs. Control): only adenine phosphoribosyltransferase significantly upregulated

DNMT-TKO 2iL (AzadC vs Control): -

WT – Lif (AzadC vs. Control):

Significantly down: dCTP pyrophosphatase 1, Multifunctional protein ADE2, Bifunctional purine biosynthesis protein ATIC, Dihydroorotate dehydrogenase (quinone, mitochondrial), Amidophosphoribosyltransferase

Significantly up: Adenylate kinase 4 (mitochondrial), Thymidylate kinase, Inosine triphosphate pyrophosphatase.

DNMT-TKO – Lif (AzadC vs. Control):

Significantly down: Hypoxanthine-guanine phosphoribosyltransferase, Thioredoxin reductase 1 (cytoplasmic), Inosine-5'-monophosphate dehydrogenase 2, UMP-CMP kinase, dCTP pyrophosphatase 1

Significantly up: Adenylate kinase isoenzyme 1, Nucleoside diphosphate kinase A, Nucleoside diphosphate kinase B

5. The authors did not explain why primed DNMT TKO ES cells are more sensitive to AzadC compared to naïve ones. This phenomenon does not necessarily support the conclusion that "the DNMT-dependent effect is highly dominant for the toxicity profile of AzadC when DNMT activity is high and DNMT-independent DNA lesions only play a subordinate role".

We agree with the Reviewer that AzadC is more cytotoxic to the DNMT-TKO in the primed state, which cannot be DNMT-dependent. We have included an γ H2AX staining for the DNMT-TKO in the supplement, where we compare 2.5 μ M and 10 μ M of AzadC in the naïve state to 0.4 μ M and 1.6 μ M in the primed state and the staining suggests that there is a bit more DNA damage at 1.6 μ M in the primed state compared to 2.5 μ M in the naïve state.

However, since the wt is much more affected in the primed state compared to the naïve state, we still think that our argument that DNMT-activity is the determining factor for the increased cell death in the primed state is valid. Whereas the effect of AzadC on wt and DNMT-TKO is within the same range in the naïve state (only slightly shifted towards higher concentrations), the effect of AzadC in the primed state is dramatic in the wt, but only moderate in the DNMT-TKO.

6. The third section of the Results contains excessive background information unrelated to the result descriptions, which leads to a lack of coherence. It includes unnecessary details, such as the introduction of a specific protein's function. Please revise and shorten this section.

We thank the Reviewer very much for pointing this out and have shortened this section now.

7. The authors observed that under 2iL conditions, there were no changes in chromatin-bound proteins and no activation of DDR in DNMT TKO cells. How does AzadC kill these cells? Please discuss.

The proteomics chromatin enrichment method captures robust changes of protein recruitment towards the chromatin. Recruitment of the DNA damage repair machinery to very few genomic loci per cell will probably not be captured as significantly different between the untreated control and the treated cells. One of our threshold criteria for significant enrichment/depletion is a log₂FC of $\geq \pm 1$. This means that proteins have to be detected with twice the intensity compared to the untreated control. This strict threshold should ensure that we do only capture processes with high confidence. Few DNA damage sites per cell can induce apoptosis and therefore it is no contradiction that DNMT-TKO cells show an increase in cell death but we cannot capture the DDR using our bulk chromatin-enrichment approach.

8. The authors observed a significantly lower incorporation of AzadC into the DNA of DNMT TKO cells compared to that of the wildtype cells when treated with AzaC. What could be the potential reasons for this difference?

As indicated by previous studies (Aimiwu et al., Blood. 2012;119(22):5229-5238; now referenced), AzaC limits its own conversion to AzadC and this inhibition might be more pronounced in the DNMT-TKO. To study why it is more pronounced in the DNMT-TKO is beyond the scope of this study, where the mESCs are used as a model to identify relevant mechanisms regarding AzadC and AzaC metabolism that can then serve as a starting point for investigations in clinically relevant models.

9. It is intriguing that Trdmt1 KO cells are the most sensitive genotype to AzaC treatment, even independent of dosage (Fig 4K). This should be discussed.

We don't know the reason for the higher sensitivity of the Trdmt1-KO towards AzaC. Experiments to reveal underlying reasons could be to investigate whether RNA stability is more affected in those cells and the effect is multiplied with AzaC. However, we think that this would be out of scope in this study. We kept the Trdmt1 data nevertheless because we think that it is an interesting finding for the community.

Minor concerns

1. What is the "non-canonical nucleobase 5-aza-cytosine"? What MoA does it correspond to?

We used this term to only refer to the nucleobase and not the whole nucleoside. However, we agree that it is confusing and thank the reviewer for pointing this out. We consistently replaced it now.

2. In Figure 2b, there is no statistical data on the survival of cells. How is it demonstrated that

there is a difference in cell viability between high and low concentrations? From the figure, the impact of various concentrations on cells appears to be similar for both wildtype (starting at 1.25) and TKO (starting at 2.5) cells.

The brightfield images of 2B are only a representative snapshot. We have presented the quantitative data on cell death that corresponds to that images in the bar graphs of the apoptosis data (used to be Figure 2I). We have arranged the figure now to show the apoptosis data side by side to the microscopy images to make it clear.

3. In Figure 2G, H2AX should be used as the loading control to quantify rH2AX. Gray analysis should be presented. The internal reference, H3, was overexposed and had too many nonspecific bands.

We haven't used H2AX as a loading control because H2AX is a histone variant that is only placed at DNA damage sites and if we had used it as a loading control, we would have used the effect we wanted to study (DNA damage) as a factor for normalization. However, we agree that the H3 signal is not ideal. Therefore, we have included the Ponceau S staining already in the initial manuscript. With Ponceau S, subtle differences in protein loading cannot be monitored but it is clear that the observed differences of the γ H2AX signal, especially between Ctrl and treated samples, cannot be explained by potentially lower amounts of protein that were loaded for the control samples. If this was the case, the Ponceau S staining would have indicated substantial differences

4. Figure 2I shows that 1.25 μ M has a similar cytotoxic effect on TKO and wt (wildtype) cells, which is contradictory to Figure 2B that shows that 1.25 μ M does not have a significant cytotoxic effect on TKO cells. Please clarify.

Figure 2I (now 2D) does not show a significant difference in apoptotic events between wt and DNMT-TKO after treatment with 1.25 μ M AzadC, but neither between wt Ctrl and treatment. That result is not contradicted by the BF microscopy images, which suggest mainly slower proliferation for the wt after treatment with 1.25 μ M AzadC, but no substantial effect on the DNMT-TKO. As suggested by Reviewer 1, we have included now the proliferation curves for all concentrations tested in the supplement, which supports this impression.

Summary:

Doubling times:	wt Ctrl	17 – 19 h
	wt 1.25 μ M AzadC	24 h
	wt 2.5 μ M AzadC	26 h
	wt 5.0 μ M AzadC	63 h
	DNMT-TKO Ctrl.	17 – 18 h
	DNMT-TKO 1.25 μ M AzadC	20 h
	DNMT-TKO 2.5 μ M AzadC	26 h
	DNMT-TKO 5.0 μ M AzadC	43 h

May 29, 2024

RE: Life Science Alliance Manuscript #LSA-2023-02437-TR

Prof. Franziska R Traube
University of Stuttgart
Institute of Biochemistry and Technical Biochemistry
Biochemistry of Cellular Biomedical Systems
Stuttgart 70569
Germany

Dear Dr. Traube,

Thank you for submitting your revised manuscript entitled "The type of DNA damage response after Decitabine treatment depends on the level of DNMT activity". We would be happy to publish your paper in Life Science Alliance pending final revisions necessary to meet our formatting guidelines.

- please be sure that the authorship listing and order is correct
- please upload all figure files as individual ones, including the supplementary figure files; all figure legends should only appear in the main manuscript file
- please add the Twitter handle of your host institute/organization as well as your own or/and one of the authors in our system
- please add your main, supplementary figure, and table legends to the main manuscript text after the references section
- remove your figures from the manuscript file
- please upload a clean manuscript file without the track changes
- please provide your table at the end of the manuscript file or upload them separately in editable Word or Excel format
- the contributions selected for Stylianos Michalakis do not qualify them for authorship. Please either update the contributions in our system and the Author Contributions section of the manuscript or let us know if the author needs to be removed.
- please consult our manuscript preparation guidelines <https://www.life-science-alliance.org/manuscript-prep> and make sure your manuscript sections are in the correct order
- please add your supplementary references to the main reference section in the manuscript text
- please add a callout for Figure S3F to your main manuscript text
- there should be a single Data Availability statement at the end of the Materials and Methods section that includes the accession info. The Pride dataset should be made publicly accessible at this point, removing the need for the Reviewer access information.
- please make sure that the callouts throughout the text for the various Supplemental Data Files indicate which File is being referred to

A. FINAL FILES:

B. MANUSCRIPT ORGANIZATION AND FORMATTING:

Sincerely,

Reviewer #1 (Comments to the Authors (Required)):

The authors have addressed my concerns.

June 11, 2024

RE: Life Science Alliance Manuscript #LSA-2023-02437-TRR

Prof. Franziska R Traube
University of Stuttgart
Institute of Biochemistry and Technical Biochemistry
Biochemistry of Cellular Biomedical Systems
Stuttgart 70569
Germany

Dear Dr. Traube,

Thank you for submitting your Research Article entitled "The type of DNA damage response after Decitabine treatment depends on the level of DNMT activity". It is a pleasure to let you know that your manuscript is now accepted for publication in Life Science Alliance. Congratulations on this interesting work.

DISTRIBUTION OF MATERIALS:

Again, congratulations on a very nice paper. I hope you found the review process to be constructive and are pleased with how the manuscript was handled editorially. We look forward to future exciting submissions from your lab.

Sincerely,
